# Locate-then-edit for Multi-hop Factual Recall under Knowledge Editing

**Zhuoran Zhang** [1 2]   **Yongxiang Li** [2 3]   **Zijian Kan** [2 3]   **Keyuan Cheng** [2 3]   **Lijie Hu** [2 4]   **Di Wang** [2 4]

## Abstract

The locate-then-edit paradigm has shown significant promise for knowledge editing (KE) in Large Language Models (LLMs). While previous methods perform well on single-hop fact recall tasks, they consistently struggle with multi-hop factual recall tasks involving newly edited knowledge. In this paper, leveraging tools in mechanistic interpretability, we first identify that in multi-hop tasks, LLMs tend to retrieve knowledge with implicit subject information from deeper MLP layers, unlike single-hop tasks, which rely on shallow layers. This distinction explains the poor performance of current methods in multi-hop queries, as they primarily focus on editing shallow layers with single-hop edit prompts, leaving deeper layers unchanged. To address this, we propose IFMET, a novel locate-then-edit KE approach designed to edit both shallow and deep MLP layers. Beyond single-hop editing prompts, IFMET further incorporates multi-hop editing prompts to locate and modify knowledge across different stages of reasoning. Experimental results demonstrate that IFMET significantly improves performance on multi-hop factual recall tasks, overcoming the limitations of previous locate-then-edit methods.

## 1. Introduction

Large Language Models (LLMs) like ChatGPT (Achiam et al., 2024) and LLaMA-2 (Touvron et al., 2023) have emerged as powerful knowledge bases, demonstrating remarkable abilities in both factual knowledge representation and reasoning over complex queries (Etezadi & Shamsfard, 2022). However, as the need for updating and correcting knowledge within these models grows, research on knowledge editing (KE) has gained significant attention, focusing on cost-effective ways to modify specific information in

LLMs (Mazzia et al., 2023). KE methods can be broadly classified into two categories based on whether they alter the original model weights: weight-preserving (Zhong et al., 2023) and weight-modifying approaches (Meng et al., 2022a;b). Weight-preserving methods aim to modify the model's outputs by integrating external memory or leveraging strategies such as in-context learning without altering the underlying weights (Cheng et al., 2024b;a). Weight-modifying methods can be further categorized into learning-based and optimization-based methods. The former update weights using gradients but face challenges such as overfitting and poor generalization. The latter, such as ROME (Meng et al., 2022a) and MEMIT (Meng et al., 2022b), have introduced the "locate-then-edit" paradigm, which first identifies the knowledge storage layers and then adjusts their weights through optimization techniques to achieve the desired knowledge modification.

Compared to weight-preserving methods and learning-based weight-modifying approaches, the locate-then-edit paradigm offers precise editing of the model's internal knowledge with low computational costs (Zhang et al., 2024). However, despite the success of locate-then-edit methods in single-hop fact recall tasks (Li et al., 2024c), they share a common limitation (Zhong et al., 2023): **The post-edited model struggles with multi-hop factual recall tasks involving the newly edited knowledge** (see Table 3 for details). For example, after changing the knowledge(fact) "The capital of Spain" from "Madrid" to "Hartford", the model correctly answers $Q_1 = $ "What is the capital city of Spain?". However, when posed with the multi-hop question $Q_2 = $ "What is the capital city of the country where Pablo Picasso holds citizenship?", it still responds with "Madrid" (Figure 1 (b)). This discrepancy raises a natural question: *Has the locate-then-edit paradigm reached its limits for multi-hop factual recall tasks, or does it still hold unexplored potential?*

To address this question, we focus on the key distinction between the existing locate-then-edit paradigm and the multi-hop fact recall task, which lies in the single-hop editing prompts. Previous methods were primarily designed for single-hop factual recall tasks. They typically associate an edit instance, such as (Spain, Capital, Madrid → Hartford), with a single-hop editing prompt "The capital of Spain is" and use it for the following editing process. Considering

[1]Peking University [2]Provable Responsible AI and Data Analytics (PRADA) Lab [3]South China University of Technology [4]King Abdullah University of Science and Technology. Correspondence to: Lijie Hu <lijie.hu@kaust.edu.sa>, Di Wang <di.wang@kaust.edu.sa>.

*Proceedings of the 42$^{nd}$ International Conference on Machine Learning*, Vancouver, Canada. PMLR 267, 2025. Copyright 2025 by the author(s).

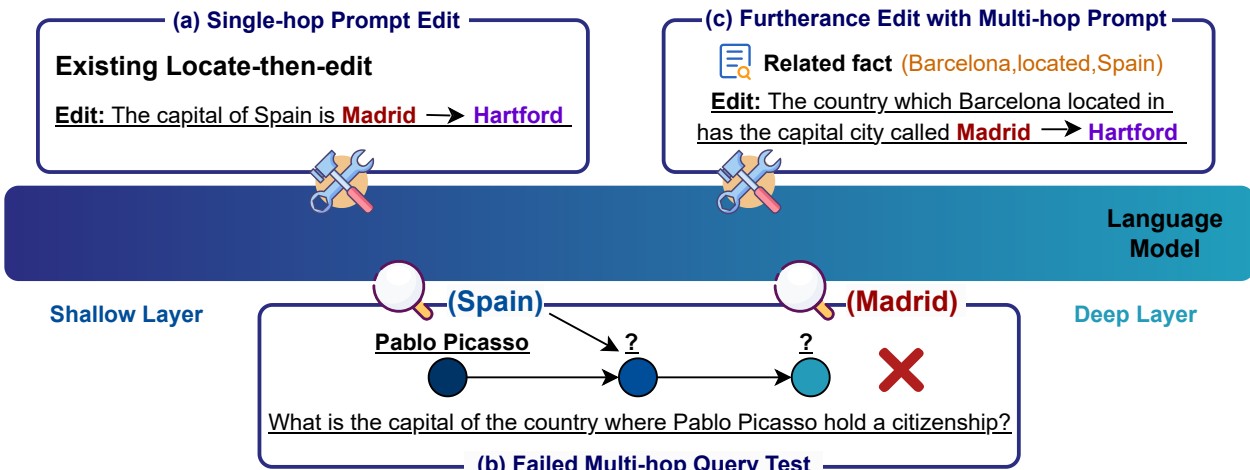

*Figure 1.* **(a)** The existing locate-then-edit KE method updates **new fact** to the shallow layers of the model using a single-hop edit prompt. **(b)** For multi-hop fact recall tasks, especially when the edited fact is in the second or subsequent hops, the hops typically access the deeper layers which outputs the **unmodified knowledge**. **(c)** Our method introduces a **prefix hop** for each single-hop edit, creating a two-hop edit prompt. We utilize this new prompt to perform a furtherance edit, targeting the deeper layers for more effective knowledge updating.

this limitation, we propose the following hypotheses: 1) The recall process of the same fact may differ mechanistically between single-hop and multi-hop scenarios; 2) This discrepancy leads to the insufficiency of knowledge updates across the model, resulting in unsatisfactory performance on multi-hop factual recall tasks.

To test **hypothesis 1**, we first explored the mechanisms of the pre-edited model when handling multi-hop and single-hop factual recall tasks. Using the example mentioned (Spain, Capital, Madrid), we attempt to illustrate how the model reasons with the implicit subject "Spain" in $Q_2$, compared to the explicit mention in $Q_1$. In Section 3.1, by interpreting the information encoded in each layer's hidden states using LogitLens (nostalgebraist, 2020; Dar et al., 2023), We find that at the last token position, the information of the implicit subject accumulates before the final answer, which is significantly different from the single-hop scenario.

We then investigate the causal influence of the implicit subject on the final answer and the mechanism by which it affects the prediction. By using causal intervention experiments (Li et al., 2024d), our results indicate that in the multi-hop scenario, the implicit subject causally guides the emergence of the final answer by retrieving relevant knowledge from the **deeper MLP layers**. This contrasts sharply with the single-hop cases (Meng et al., 2022a; 2023), where the subject information is used to retrieve information from **shallow MLP layers**. Based on this difference, We provide a more detailed explanation for **hypothesis 2**: Previous methods leveraging single-hop prompts for editing are insufficient as they only update the relevant knowledge in the shallow MLP layers but fail to propagate the changes to deeper layers. **As a result, the deeper layers retain unedited knowledge that is only activated by**

**implicit multi-hop fact recall mechanisms.** Based on these observations, we developed an advanced locate-then-edit KE method specifically designed to modify knowledge in both shallow and deep MLP layers, which we named **I**nterpretability-Guided **F**urtherance **M**odel **E**diting in a **T**ransformer (**IFMET**). To surpass the limitations of single-hop prompts, IFMET generates relevant multi-hop editing prompts for each edit instance. To address the issue of insufficient knowledge updates caused by differences in the reasoning mechanisms, IFMET extends existing methods by using multi-hop prompts for furtherance editing, effectively addressing cases in the single-hop and multi-hop scenario, as illustrated in Figure 1. Our contributions can be summarized as follows [1]:

- We first identified key differences in the mechanisms the model uses for reasoning in single-hop versus multi-hop fact recall tasks. In multi-hop scenarios, unlike single-hop cases, the model prioritizes inferring the implicit subject at the last token position, which guides the generation of the final answer.

- Next, we pinpointed the components of the implicit subject that influenced the final answer within the deeper MLP layers. We demonstrated that the absence of edited knowledge of these components significantly impacted the model's performance.

- We propose IFMET, an advanced locate-then-edit KE method specifically designed to modify knowledge in both shallow and deep MLP layers using single and multi-hop edit prompts. Experimental results confirm the effectiveness of our method, showing that it suc-

---

[1]Due to the space limit, we refer readers to Appendix A for previous work.

cessfully overcomes the limitations of previous methods in handling multi-hop factual recall tasks.

## 2. Preliminaries

**Notations.** We define the set of knowledge(fact) as $\mathcal{K} = \{(s, r, o)\} \subseteq \mathcal{E} \times \mathcal{R} \times \mathcal{E}$, where $\mathcal{E}$ and $\mathcal{R}$ denote the set of entities and relations respectively. Each tuple $(s, r, o) \in \mathcal{K}$ represents that the corresponding entity of subject entity $s$ under relation $r$ is object entity $o$. An editing instance can be described in the form of a triplet: $e = (s, r, o \rightarrow o^*)$, where $o^*$ denotes the new edited object in place of the original object $o$ related to $s$ through $r$.

### 2.1. Factual Recall Tasks

**Format of Factual Recall Tasks.** Factual recall tasks refer to verifying whether the model $\mathcal{M}$ can correctly provide the final answer to a single-hop or multi-hop factual recall $Q$. $Q$ requires multi-step($\geq 1$) reasoning to reach the final answer. Its reasoning process is composed of a chain of knowledge $C = (s_1, r_1, o_1) \oplus \cdots \oplus (s_n, r_n, o_n)$, where $s_1$ is the start subject that is explicitly given in the question, $o_n$ is the final answer. There are two different format question prompts for factual recall tasks: Cloze-Format $Q_{cloze}$ and QA-Format $Q_{qa}$. For instance, given two-hop questions with the knowledge chain like *(Paradiso, author, Dante Alighieri) $\oplus$ (Dante Alighieri, country of citizenship, Italy)*, $Q_{cloze}$ can be "The author of Paradiso is a citizen of", while $Q_{qa}$ is "What country does the author of Paradiso hold citizenship in?". For better clarity, we categorize the multi-hop fact recall into two types: *explicit recall step* $(s_1, r_1, o_1)$ and *implicit recall steps* $\{(s_2, r_2, o_2), \ldots, (s_n, r_n, o_n)\}$. If the model's final answer is the same as the answer to the question, the recall is considered successful, which can be represented as $\mathcal{M}(Q_{cloze}) = o_n$ or $\mathcal{M}(Q_{qa}) = o_n$.

**Multi-hop Factual Recall under Knowledge Editing.** This task assesses whether the post-edited model can effectively leverage the updated knowledge for reasoning in multi-hop fact recall tasks. Given an edit $e = (s, r, o \rightarrow o^*)$, the edit prompt $T_e$ and a chain of facts $C_e$ which includes $(s, r, o)$ as one of its components. The post-edited model must leverage the new factual knowledge $(s, r, o^*)$ to answer the multi-hop query. For example, given edit *(Paradiso, author, Dante Alighieri $\rightarrow$ Mark Twain)*, the model's response of "The author of Paradiso is a citizen of" should change from the original answer *Italy* to the new answer *USA*.

### 2.2. Mechanistic Interpretation Tools

**LogitLens.** LogitLens (nostalgebraist, 2020) is a framework for interpreting the hidden states (activations) of language models such as GPT (Brown et al., 2020). For the hidden state $h_l^i$ (token $i$ at the $l$-th layer), the logits $s_l^i$ and

probabilities $p_l^i$ over the output vocabulary set $V$ are:

$$\begin{cases} s_l^i = W_U h_l^i \in \mathbb{R}^{|V|}, \\ p_l^i = \text{softmax}\left(s_l^i\right) \end{cases}$$

where $W_U$ denotes the unembedding matrix used in the final prediction layer of the model. LogitLens also works for the decomposition of hidden states, such as MLPs $m_l^i$ and attention heads $a_l^i$, where $h_l^i = h_{l-1}^i + m_l^i + a_l^i$. [2] LogitLens posits that probabilities and logits provide insights into how the model prioritizes different potential tokens, as indicated by the proportion of related information. So we define $\text{Info}(h_l^i, j)$ as the information related to token $j \in V$ contained in $h_l^i$, positively correlated with $s_l^i[j]$ and $p_l^i[j]$. To account for the probability variations across different layers, we define $\text{Info}(h_l^i, j)$ as the layer-wise min-max normalized probability (Li et al., 2024d), where $L$ is the total number of layers:

$$\begin{cases} p_{max}^i[j] = \max_{\{l=1,\ldots,L\}} p_l^i[j], \\ p_{min}^i[j] = \min_{\{l=1,\ldots,L\}} p_l^i[j], \\ \text{Info}(h_l^i, j) = \frac{p_l^i[j] - p_{min}^i[j]}{p_{max}^i[j] - p_{min}^i[j]} \end{cases}$$

**Causal Intervention on Hidden States.** Causal intervention on hidden states (Li et al., 2024d;a) involves deliberately altering specific hidden states in a model to observe the resulting changes in various metrics, thereby helping to establish cause-and-effect relationships. This process includes three pivotal components: the intervention operation $\mathcal{I}$ to be conducted, the target hidden state or its decomposition $\mathcal{H}$ selected for intervention, and the effect metric *IE* which measures the change caused by the intervention $\mathcal{I}$.

## 3. Mechanisms of Knowledge Storage and Reasoning

In this section, we will explore the reasoning mechanisms of the pre-edited model for multi-hop factual recall tasks and how they differ from those in single-hop. Specifically, we focus on two-hop tasks to better illustrate these distinctions, whose knowledge chain is represented as $C = (s_1, r_1, o_1) \oplus (s_2, r_2, o_2)$. In Section 3.1, we primarily investigate the differences in the recall mechanism of the same fact $(s_2, r_2, o_2)$ when it severs as the implicit multi-hop step in the chain and the explicit single-hop reasoning step. In Sections 3.2, we explain why the model edited by existing method tends to output the original answer instead of the new edited one in multi-hop scenario[3].

---

[2]We employ GPT variants such as GPT-J (Wang & Komatsuzaki, 2021) that position attention in parallel to the MLP, which mathematically equates to models that calculate MLP sequentially

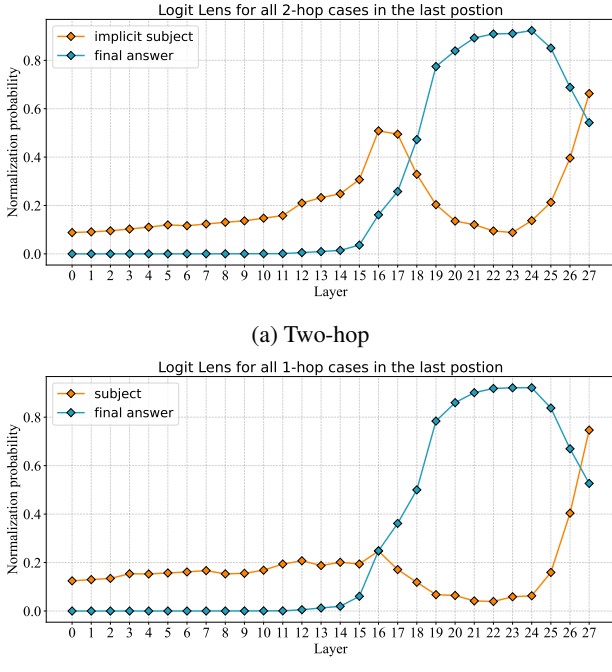

(a) Two-hop

(b) Single-hop

*Figure 2.* **LogitLens results of the last token position at different layers.** Yellow line represents the information containing implicit subject $s_2$, i.e., Info($h_l, s_2$). Blue line represents the information for the final answer, i.e., Info($h_l, o_2$). Larger versions of the sub-figures are available in Appendix Figure 6.

### 3.1. How Pre-edited Models Reason Fact Recall Tasks

For a multi-hop fact recall task, the model may employ multiple strategies to answer such tasks, including the formation of a single super-relation (Ju et al., 2024) $(s_1, r_{mul}, o_n)$, where $r_{mul} = r_1 \rightarrow \cdots \rightarrow r_n$, or by segmenting the task into one explicit recall step followed by several implicit recall steps to answer step-by-step. Previous research (Hou et al., 2023) suggests that models typically engage in reasoning by considering each single-hop recall individually.

Based on this, we hypothesize that the model prioritizes deducing the implicit subject $s_2$ and subsequently recalls the final answer $o_2$ based on it. We conduct exploratory experiments about the role of layer hidden state and components like MLP and Attention head at different positions of prompt. The following sections aim to verify this hypothesis by addressing the three questions: **Q1**: In the case of multi-hop reasoning, is the information related to $s_2$ accumulated before that of $o_2$? **Q2**: Does the accumulation of relevant information about $s_2$ causally influence the model's reason-

---

after the attention module, as discussed in (Brown et al., 2020).

[3]All experiments in section 3 are conducted using a subset of single and two-hop data from MQuAKE-CF (Zhong et al., 2023) with the GPT-J (6B) model (Wang & Komatsuzaki, 2021). More detailed information about the data and the experimental setup is provided in Appendix B.2.1.

ing for $o_2$? **Q3**: Which component facilitates the influence of factual recall process from $s_2$ to $o_2$?

**Is the information related to $s_2$ accumulated before that of $o_2$ ?** We use LogitLens to examine the accumulation of information related to the implicit subject $s_2$ and the final answer $o_2$ in the two-hop scenario. The model's predictions for $o_2$, are derived from the last token of the prompt, where crucial information about the resolved implicit subject $s_2$ should be propagated (Biran et al., 2024). Therefore, we focus on the hidden state $h_l$ at the $l$-th layer of the last token position, analyzing Info($h_l, s_2$) and Info($h_l, o_2$) as measures of the information related to $s_2$ and $o_2$ contained in $h_l$.

The results, depicted in Figure 2a, show that Info($h_l, s_2$) gradually reaches its peak during middle layers [15-17], while Info($h_l, o_2$) increases and peaks during later layers [21-24]. This pattern suggests that, in multi-hop tasks, the implicit subject $s_2$ is processed during the middle layers before reaching the final answer $o_2$. We conducted a similar experiment by giving $s_2$ explicitly in a single-hop prompt. The results, shown in Figure 2b, indicate that there is no significant peak for the subject information before the final answer probability begins to accumulate, suggesting that in single-hop cases, the accumulation process of the final answer at the last token is not significantly correlated with the subject information.

> **Takeaway 1**
>
> In multi-hop scenarios, the implicit subject information consistently accumulates before the final answer at the last token position. However, in single-hop scenarios, since the subject is explicitly given, there is no need for accumulation at the last token position.

**Does the accumulation of relevant information about $s_2$ causally influence the model's reasoning for $o_2$?** We propose an intervention experiment where we reduce the information content of $s_2$ at the subject token and last token position, then observe changes in the output probability of the final answer in the last prediction layer.

Specifically, we replace the hidden state $h_l$ (in layer $\ell$ of the last token or subject token) with $h_l^*$, and the corresponding logits $s_l \, (= W_U h_l)$ changes to $s_l^* \, (= W_U h_l^*)$. $s_l^*$ is defined as:

$$s_l^*[j] = \begin{cases} \min(s_l[j]), & \text{if } j \in s_2 \\ s_l[j], & \text{otherwise,} \end{cases} \quad (1)$$

where we minimize the logits corresponding to the tokens in $s_2$ without altering the logits of other tokens, aiming to diminish the effect of $s_2$. Follow the setting in (Meng et al., 2022a), we select the window size = 5, meaning that in each intervention we consider the hidden states across five consecutive layers centered on the targeted layer. This setup

allows us to describe the process through a causal intervention framework, where the target $\mathcal{H}$ is $h_l$, the intervention $\mathcal{I}_h$ and the effect $IE_h$ are defined as follows:

$$\mathcal{I}_h : h_l^* = h_l + \arg\min_{\Delta h_l} \|W_U(h_l + \Delta h_l) - s_l^*\|^2,$$

$$IE_h = p_L[j] - p_L^E[j], \quad j \in o_2, \qquad (2)$$

where $L$ is the last layer, $p_L[j]$ denotes the original output probability of the first token of $o_2$ in the $L$-th layer, and $p_L^E[j]$ is the probability after the intervention is applied. This approach illustrates how the hidden states and probabilities are expected to change when the logits are modified to $s^*$. For computational efficiency, we opt to approximate $h_l^*$ using a combination of least squares and minimum-norm methods (Lawson & Hanson, 1995) (further details are provided in Appendix C.1).

Figure 3a presents the outcomes of our intervention experiments across all layers, where a brighter color signifies a stronger intervention effect. We found a clear positive impact from intervening in layers [17-18] for the last token group. In comparison, the subject token group demonstrated only slight influences, confined mainly to the early layers. This suggests that, in the last token position, the information of $s_2$ encoded in the intermediate layers plays a crucial role in the probability accumulation process of $o_2$. We attribute the significant effects observed in the last two layers to the model's tendency to increase the probability of output articles in these layers, thereby influencing the likelihood of answer generation, rather than involving factual recall mechanisms.

> **Takeaway 2**
>
> Unlike the mechanism of reasoning the knowledge in single-hop scenarios, in the reasoning process of the second-hop knowledge in two-hop scenarios, **the accumulated implicit subject information has causal effects on the final answer**.

**Which component facilitates the influence of the factual recall process from $s_2$ to $o_2$?** Previous studies claimed that single-hop tasks using subject information to retrieve knowledge from MLP layers (Meng et al., 2022a;b). By comparing the effects of MLP and attention components, we seek to answer how the implicit subject $s_2$ influences the prediction of the final answer $o_2$ in the multi-hop scenario. We conducted causal intervention experiments similar to the experiments above but focused specifically on the Attention Head and MLP components. Specifically, we aim to replace $m_l$ (the input hidden state of the last token in the $l$-th MLP or attention head) with $m_l^*$, where we have $s_l = W_U m_l$ and $s_l^* = W_U m_l^*$ with $s_l^*$ is same as in (1). The intervention $\mathcal{I}_m$ shares the same idea as in (2), except that $h_l$ is replaced with $m_l$. We follow the definition of the intervention effect $IE_h$, which is the probability change calculated from the

last layer $L$. In total, our causal intervention is formulated as

$$\mathcal{I}_m : m_l^* = m_l + \arg\min_{\Delta m_l} \|W_U(m_l + \Delta m_l) - s_l^*\|^2,$$

$$IE_h = p_L[j] - p_L^E[j], \quad j \in o_2.$$

Figure 3b presents the outcomes of our intervention experiments across all layers. The clear positive impact from intervening in the intermediate layers [19-25] is demonstrated in the MLP group, in contrast to negligible effects observed in the attention head group in the deeper layers. This suggests that the implicit subject $s_2$ at the last token position was used for retrieving the related information of $o_2$ from **deeper MLP layers**. Note that previous work (Meng et al., 2022a; 2023) has mentioned that explicit single-hop tasks primarily rely on the subject token position to retrieve information from **shallow MLP layers**.

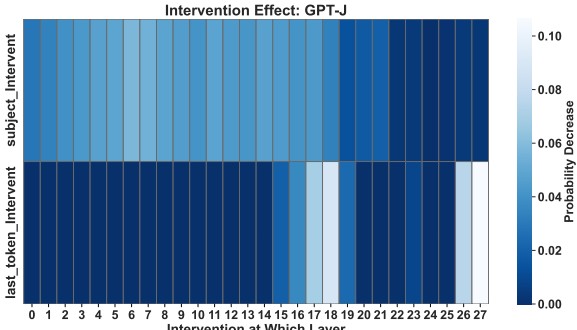

(a) Causal intervention results of layer hidden state at the subject token and last token position.

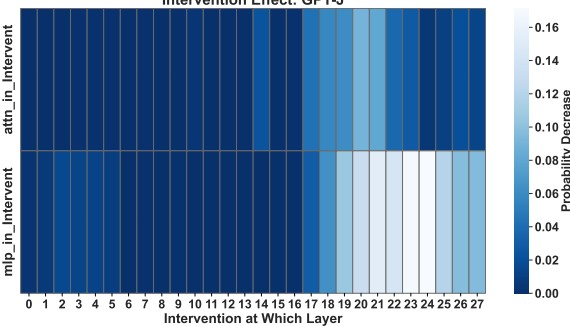

(b) Causal Intervention result of hidden state from Attention head and MLP at last token position.

*Figure 3.* **Causal Intervention Result**: A brighter color signifies a stronger intervention effect. Note that negative effect values ($\leq 0$) are clipped to 0 in both groups for better visualization. (a) is probability change $IE_h$ of intervention $\mathcal{I}_h$, (b) is probability change $IE_h$ of intervention $\mathcal{I}_m$.

**Takeaway 3**

In single-hop fact recall, relevant knowledge is retrieved through the token at the subject position, utilizing **shallow MLP layers**. In contrast, when the same knowledge serves as an implicit reasoning step in multi-hop, it is retrieved through the token at the last position, utilizing **deeper MLP layers**.

### 3.2. Why Existing locate-then-edit KE Methods Failed

Based on the findings above, we can explain the unsatisfactory performance of the existing locate-then-edit methods. The factual recall mechanism for the same knowledge differs when it serves as a single-hop reasoning step versus an implicit reasoning step in multi-hop reasoning. Consequently, for an editing instance $(s, r, o \to o^*)$, using only the corresponding explicit single-hop prompt for editing in previous methods is insufficient as they only update the relevant knowledge in the shallow MLP layers but fail to propagate the changes to deeper layers, which are essential for multi-hop factual recall tasks.

We provide a concrete example in Table 1 for a better understanding. Given an editing instance $e$ and $T_e$. Existing methods modify the weights of shallow MLPs with $T_e$ to make it answer *Hartford*. $C_{pre}$ and $C_{post}$ represent the multi-hop factual recall chains obtained by $e$ as an explicit recall step and an implicit recall step, respectively. $T_{C_{pre}}$ and $T_{Cpost}$ represent the corresponding prompts. In this example, the query $T_{C_{pre}}$ should be answered correctly because the explicit fact $(Spain, capital, Hartford)$ can be recalled in shallow MLPs. However, the $T_{C_{post}}$ is still answered with *Madrid* because the knowledge $(Spain, capital, Madrid)$ stored in deeper MLPs does not change. To verify our claim above, we divide the two-hop factual recall tasks into two sets, $D_{Pre}$ and $D_{Post}$, depending on the position of the edited knowledge within the two-hop reasoning process. Specifically, for an edit instance $e = (s, r, o, o^*)$, we have the following two sets:

$$D_{Pre} = \{(s, r, o^*) \oplus (s_2, r_2, o_2)\},$$
$$D_{Post} = \{(s_1, r_1, o_1) \oplus (s, r, o^*)\}.$$

The specific experimental details are in the appendix C.2. Table 2 presents the results of the comparative experiments. As shown, the multi-hop question accuracy performance of the existing SOTA locate-then-edit method PMET on $D_{Pre}$ is significantly better than on $D_{Post}$, which aligns with our expectations. This is because the reasoning in the explicit recall step is similar to the single-hop process. After updating the knowledge in the shallow MLP layers by single-hop edit prompt, the newly edited knowledge can be leveraged in $D_{pre}$. In contrast, for cases in $D_{Post}$, the model fails to produce the correct final answer because existing methods didn't update the knowledge in deeper MLP layers.

| | |
|---|---|
| $e$ | (Spain, capital, Madrid $\to$ Hartford) |
| $T_e$ | The capital city of Spain is |
| $C_{pre}$ | e $\oplus$ (Hartford, mayor, Arulampalam) |
| $T_{C_{pre}}$ | The mayor of the capital city of Spain is |
| $C_{post}$ | (Barcelona, country, Spain) $\oplus$ e |
| $T_{C_{post}}$ | The capital city of the country where Barcelona is located is |

Table 1. An example for an single-hop edit instance and its corresponding multi-hop prompt.

Table 2. Comparison of multi-hop Acc for $D_{Pre}$,$D_{Post}$.

| Edit Method | QA Format(%) ↑ | | Cloze Format(%) ↑ | |
|---|---|---|---|---|
| | $D_{Pre}$ | $D_{Post}$ | $D_{Pre}$ | $D_{Post}$ |
| Base | 50.62 | 41.72 | 20.31 | 18.63 |
| PMET | **64.29** | **2.93** | **43.37** | **4.60** |

## 4. IFMET

Motivated by our findings on the distinctions between single-hop and multi-hop factual recall processes, we introduce the **I**nterpretability-Guided **F**urtherance **M**odel **E**diting in a **T**ransformer (**IFMET**). IFMET extends the existing locate-then-edit paradigm in two ways: first, constructing multi-hop edit prompts for each edit instance to expand the original single-hop edit prompt, and second, adding a furtherance editing step that applies multi-hop edit prompts to deeper MLPs. IFMET thoroughly integrates new knowledge across shallow and deeper MLP layers, significantly improving the model's accuracy and robustness in multi-hop factual recall scenarios.

**Multi-hop edit prompt construction.** For a given edit $e = (s, r, o \to o^*)$, existing locate-then-edit methods provide only a single-hop edit prompt such as $T_e$ in Table 1. Through Section 3.2, we recognize that the main limitation of these methods lies in ignoring the difference between explicit single-hop and implicit multi-hop reasoning mechanisms. Single-hop edit prompts only modify knowledge in the shallow MLP layers, leading to poor performance in the $C_{post}$ format multi-hop factual recall. Therefore, we aim to construct $T_{Cpost}$-format multi-hop edit prompt for each edit instance, just like mentioned in Table 1. We first transform each edit instance into a two-hop fact recall chain $C = (s', r', o') \oplus (s, r, o)$ where $o' = s$, then transform this two-hop factual recall chain into an edit prompt by using the corresponding prompt templates. The key step is to identify the relevant preceding knowledge $(s', r', s)$ for $e$, which can be sourced from any valid knowledge base. Here, we provide two methods: one based on WikiData and the other using the model itself. Further discussion of these two construction methods is detailed in Appendix D.1.

**IFMET.** Now we introduce the proposed IFMET frame-

| Editor +CoT | Average | # Edits = | | | | # hops = | | |
|---|---|---|---|---|---|---|---|---|
| | | 1-edit | 2-edit | 3-edit | 4-edit | 2-hop | 3-hop | 4-hop |
| Base | 42.83 | 36.96 | 45.27 | 46.85 | 48.51 | 48.9 | 30.7 | 48.9 |
| FT | 1.9 | 4.2 | 0.7 | 0.3 | 0.0 | 3.7 | 1.4 | 0.5 |
| MEND | 11.5 | 16.0 | 11.0 | 7.3 | 4.4 | 13.9 | 11.3 | 9.5 |
| ROME | 18.1 | 23.8 | 20.9 | 9.0 | 2.6 | 33.8 | 9.1 | 11.4 |
| MEMIT | 12.3 | 20.5 | 9.8 | 5.5 | 2.6 | 22.5 | 6.0 | 8.4 |
| PMET | 17.04 | 22.63 | 16.74 | 11.19 | 7.84 | 26.65 | 12.76 | 11.7 |
| **IFMET (ours)** | **31.01** | **30.26** | **35.21** | **24.30** | **31.72** | **44.06** | **23.58** | **25.4** |

*Table 3.* **Multi-hop accuracy** comparison of different methods on the MQuAKE-3K dataset in a few-shot setting, showing the **Base** model's performance on the unedited answer and the edited model's performance on the edited answer.

work, providing a single-hop edit prompt and multi-hop edit prompt for each edit instance. Based on the difference between the single and multi-top reasoning mechanisms we discussed above, in the first edit stage, we use the single-hop edit prompt to edit shallow MLPs. In the second stage, we further use the multi-hop edit prompt to edit deeper MLPs.

Based on previous key-value memories (Geva et al., 2021), our method to edit the MLP is based on the hypothesis that factual knowledge is stored within the Feedforward Neural Networks (FFNs) of MLPs. Specifically, for the $l$-th layer FFN, its output of the $i$-th token's hidden state $h_{l-1}^i$, is given by: $v_l^i = f(W_l^{in} h_{l-1}^i) W_l^{out}$, where $f(\cdot)$ is the activation function, and $h_{l-1}^i$ is the input of the $l$-th MLP layer (for simplicity, the superscript $l$ is omitted in the following discussion). In this context, $f(W^{in} h^i)$ functions as the keys, denoted as $k_i$, the outputs represent the corresponding values $v_i$, and $W^{out}$ denotes the weights of the knowledge stored in the FFN that needs modifying. Such a structure is well aligned with the triplet form $(s, r, o)$, where the keys $k_i$ correspond to entities of interest $s_i$ or some specific fact $(s_i, r_i)$ and values $v_i$ contain information about $o_i$. Thus, we have $W^{out} k = v$ for $(k, v)$, which represents the fact $(s, r, o)$ (Geva et al., 2021). We aim to modify $W^{out}$ such that $W^{out} k = v^*$, where $v^*$ contains the information of the new knowledge.

Motivated by the above, in IFMET, there are two main steps for both the first and second edit stages: $Search$ and $Calculate$. The $Search$ process identifies the suitable $v^*$ through the edit prompt. Then the $Calculate$ process computes the change in weights $W^{out}$ using $v^*$. These two processes are foundational in existing knowledge editing methodologies. In experiments, we adopt the state-of-the-art locate-then-edit method PMET (Li et al., 2024c). The primary differences between the first and further edit stages are reflected in the used edit prompt and the layers edited. Specifically, for the edit instance $e = (s, r, o \rightarrow o*)$, the first edit utilized a single-hop edit prompt $T_e$ provided by the dataset to edit shallow layers of the model. For the furtherance edit, the two-hop prompt $T_{C_{post}}$ composed of

$(s', r, s)$ and $(s, r, o^*)$ was used, and this prompt was applied to edit deeper layers of the model. Due to space limitations, the flowchart of the algorithm and related implementation details are provided in Algorithm 1 and Appendix D.2.

## 5. Experiments

### 5.1. Experimental Setup

**Dataset and Baselines**[4]. MQuAKE-3K (Zhong et al., 2023), a challenging and widely used dataset designed to evaluate models' ability to perform multi-hop fact recall with newly edited knowledge. Each edit instance contains a multi-hop factual recall chain and the corresponding multi-hop textual question. The instance includes at least one fact from the chain to be edited and provides its single-hop edit prompt. Baselines are **Base**, which refers to the original GPT-J(6B) model without any edits; **FT** basic fine-tuning method; **MEND** (Mitchell et al., 2022), which employs meta-learning for weight updating; **ROME** (Meng et al., 2022a), the classic single-stage locate-then-edit method; **MEMIT** (Meng et al., 2023), which extends ROME to edit a large set of facts by updating weights in a range of layers; **PMET**, locate-then-edit method with FFN optimization.

**Setup and Hyperparameters.** To evaluate the performance of different KE methods, we adopt Multi-hop question answering accuracy(Multi-hop Acc) as the primary metric. For each query, the **unedited answer** denotes the expected old fact before knowledge editing, while the **edited answer** represents the expected new fact after editing. We use PMET as our primary experimental method for both the first and furtherance edits and construct multi-hop edit prompts from the knowledge triples of MQuAKE-3K to support our **IFMET**. Additional details are presented in Appendix E.2.

[4]More details about the dataset in Appendix B.1. We mainly compare IFMET with previous weight-modifying approaches, especially these single-stage edit methods applying shallow MLP edits based on single-hop edit prompts. Comparison with weight-preserving methods is discussed in section G.1

| Editor | Average↑ | Pre ↑ | Post ↑ |
|---|---|---|---|
| Base | 7.70 | 11.97 | 6.99 |
| Base+CoT | 6.83 | 8.22 | 6.60 |
| PMET | 11.17 | 39.20 | 6.53 |
| PMET+CoT | 17.04 | 43.66 | 12.63 |
| IFMET | 23.04 | 38.03 | 20.55 |
| **IFMET+CoT** | **31.01** | **43.66** | **28.90** |

(a) Edited Answer

| Editor | Average ↓ | Pre ↓ | Post ↓ |
|---|---|---|---|
| Base | 39.63 | 34.04 | 40.56 |
| Base+CoT | 42.83 | 41.31 | 43.08 |
| PMET | 29.95 | **10.09** | 33.26 |
| PMET+CoT | 29.35 | 12.67 | 32.13 |
| IFMET | 23.08 | 11.27 | 25.02 |
| **IFMET+CoT** | **21.32** | 10.80 | **23.08** |

(b) Unedited Answer

*Table 4.* Multi-hop accuracy comparison for edited and unedited answers using PMET and our editors on the MQuAKE-3K dataset. Average accuracy is calculated as the weighted average of results from these two categories, which have respective quantities of 426 and 2574. Additionally, +CoT denoted the performance incorporating a Chain-of-thought (CoT) prompt.

## 5.2. Experimental Results

**General performance.** Table 3 demonstrates the performance of various established methods alongside **IFMET** on MQuAKE-3K. To thoroughly explore the model's ability to leverage the newly edited knowledge, we use Chain-of-Thought (CoT) prompting to guide the model's responses to multi-hop tasks in this experiment. **# Hops** refers to the number of hops of the multi-hop fact chain in the edit instance, with a maximum of 4 hops. **# Edits** quantifies how many individual facts within the chain are edited, and its maximum value is the same as the maximum number of hops in the instance. What can be observed is that, on the overall average performance, IFMET consistently outperforms

| Editor | Multi-hop ↑ | Efficacy ↑ |
|---|---|---|
| IFMET | 28.38 (↑**78.0%**) | 99.56 (↑12.8%) |
| w/o $First$ | 23.14 (↑45.1%) | 66.59 (↓**24.6%**) |
| w/o $Multi$ | 17.69 (↑10.9%) | 100.00 (↑**13.3%**) |
| w/o $Deeper$ | 15.07 (↓**5.4%**) | 99.56 (↑12.8%) |
| PMET | 15.94 | 88.21 |

*Table 5.* The results of the ablation experiments on GPT-J-6B model using a subset of MQuAKE-CF. Both the percentages of decrease(↓) and increase(↑) are calculated relative to **PMET** as the baseline. The most significant performance decline is highlighted in **red** and the most significant performance increase is highlighted in **green**.

previous methods by a significant margin. Moreover, the performance improvement of IFMET is consistent across all subsets (e.g., 2-edit or 2-hop) of the dataset. Notably, in more complex reasoning scenarios, such as when edits>2 or hops>3, IFMET achieves a performance improvement of two to three times. This demonstrates the adequacy of knowledge updates in the IFMET method.

**Comparison between the unedited answer and edited answer.** In Section 3.2, we attempted to explain and demonstrate why existing methods still tend to output the unedited answer in multi-hop tasks, especially in the post-type multi-hop fact recall task. Here, we provide a more detailed breakdown of the dataset to investigate whether IFMET effectively alleviates this issue. Please refer to Appendix E.1 for the classification of {Pre,Post}. Results in Table 4 show that, in the Post scenarios where the facts are typically treated as implicit reasoning steps, IFMET effectively reduces the output of the unedited answer and improves the accuracy of the correct answer. On the pre-type tasks, as expected, existing methods demonstrate performance comparable to IFMET. This is because they primarily modify knowledge in the shallower layers.

**Ablation study.** With **Efficacy** metric, which measures whether the model can successfully answer the single-hop fact recall prompt, we comprehensively evaluate the model's ability to perform both single-hop and multi-hop reasoning using the edited knowledge. The importance of each component is reflected through comparisons of performance improvements over PMET. From the analysis of the ablation experiments in Table 5, we derive the following conclusions: **w/o $First$**: Only modifying the deeper layers using second stage edit with multi-hop edit prompt effectively enhances performance on multi-hop reasoning tasks. However, the absence of the single-hop edit prompt in the first stage resulted in the shallow MLP layers not being updated, leading to poor performance in single-hop fact recall tasks. This highlights the importance of the two-stage editing process. **w/o $Multi$**: In the second editing stage, we try to use the original single-hop edit prompt instead of a multi-hop edit prompt to edit the deeper MLP layers. However, the results corroborate our interpretability analysis which emphasizes the differences between single-hop and multi-hop reasoning mechanisms. Single-hop prompts cannot correctly modify knowledge in deep MLPs, highlighting the critical importance of multi-hop prompts. **w/o $Deeper$**: In this setup, we try to use the multi-hop edit prompt to edit shallow MLP layers (rather than deeper MLP layers). As observed across the table, there was a consistent minor fluctuation in performance. In contrast to **IFMET**'s +70% improvement, this underscores the necessity of editing knowledge in the deeper MLP layers when using the multi-hop prompts. More comprehensive ablation studies and discussions can be found in Appendix F.

**Generalizability.** Additionally, we conducted experiments on larger edit batches, newer models and more metrics in Appendix G, all of which showed significant performance improvements, effectively demonstrating the generalizability of our approach.

## 6. Conclusion

We focused on developing locate-then-edit knowledge editing methods for multi-hop factual recall tasks. We first verified that in multi-hop tasks, LLMs tend to retrieve implicit subject knowledge from deeper MLP layers, unlike single-hop tasks, which rely on earlier layers. This distinction explains the poor performance of current methods in multi-hop queries, as they primarily focus on editing shallow layers, leaving deeper layers unchanged. We then proposed IFMET, a novel locate-then-edit KE approach designed to edit both shallow and deep MLP layers. Experimental results demonstrate that IFMET significantly improves performance on multi-hop factual recall tasks.

## Impact Statement

This paper presents work whose goal is to advance the field of knowledge editing. We aim to enhance the locate-then-edit paradigm to address multi-hop factual recall tasks. There are many potential societal consequences of our work, none of which we feel must be specifically highlighted here.

## Acknowledgements

This work is supported in part by the funding BAS/1/1689-01-01, URF/1/4663-01-01, REI/1/5232-01-01, REI/1/5332-01-01, and URF/1/5508-01-01 from KAUST, and funding from KAUST - Center of Excellence for Generative AI, under award number 5940.

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

# A. Related Work

**Parameter-based Editing** Knowledge editing refers to modifying outdated, inaccurate, or harmful knowledge in LLMs without the need for retraining. Parameter-editing methods achieve this by adjusting the model's internal parameters to update its knowledge while ensuring that information unrelated to the editing domain remains unaffected. An example is ROME (Meng et al., 2022a), which explored the knowledge storage mechanisms in single-hop factual recall tasks based on causal tracing methods and proposed the Rank-One Model Editing method. Together with KN (Dai et al., 2022), it pioneered a paradigm of locate-then-edit, providing guidance for subsequent editing methods. The later extended versions, MEMIT (Meng et al., 2023), MALMEN (Tan et al., 2023), and EMMET (Gupta et al., 2024), further improved ROME by addressing its limitations in large-scale editing, enabling comprehensive edits in a single operation while demonstrating exceptional performance. Meanwhile, PMET (Li et al., 2024c) achieved more precise model editing by decoupling the residual flow of the Transformer into three components: Multi-Head Self-Attention (MHSA), Feed-Forward Networks (FFN), and residual connections, utilizing only the optimized hidden states of the FFN to accurately update FFN weights. Additionally, MEND (Mitchell et al., 2022) trained a hypernetwork to efficiently predict LLM weight updates, enabling rapid knowledge editing. METO (Yin et al., 2024) optimized the model's temporal prediction of facts, editing both historical and new knowledge to reduce forgetting during updates. Wilke (Hu et al., 2024) selected the layers in LLMs that best matched the knowledge pattern for editing, achieving continuous updates and corrections in the model's knowledge. Hewitt et al. (2024) used canonical examples to guide the model editing process, enabling fine-tuned adjustments to model behavior. However, these editing methods primarily focus on knowledge updates in specific layers and lack in-depth optimization for knowledge integration and application in multi-hop reasoning, rendering them inadequate for multi-hop questions. In contrast, IFMET enhances model interpretability, guiding more accurate knowledge integration and thereby improving model performance in multi-hop factual recall tasks.

**Mechanistic Interpretability** LLMs are capable of producing high-quality answers, but their internal workings remain opaque. As a result, the interpretability of LLMs has emerged as both a research hotspot and a critical area of focus. Mechanistic Interpretability refers to the effort to explain the internal mechanisms, decision-making processes, and outputs of LLMs. There are two primary approaches for interpreting large language models (LLMs) in the vocabulary space by examining hidden representations: Probing Classifiers (Belinkov & Glass, 2019; Belinkov, 2022; Wang et al., 2024) and Projecting Representations to the Vocabulary Space (Dar et al., 2022; Merullo et al., 2023; Belrose et al., 2023; Langedijk et al., 2023). The former identifies which parts of the model are crucial for specific tasks by training classifiers, known as probes, on hidden representations, while the latter involves mapping intermediate layer representations to the output vocabulary space and analyzing how these projections predict the next word. In this paper, we focus primarily on Projecting Representations. Logit Lens (nostalgebraist, 2020) extracted outputs corresponding to each layer in the decoding space by applying unembedding operations on the intermediate layers of LLMs. Geva et al. (2022) analyzed the nature of updates at each layer by comparing differences in logit outputs. Merullo et al. (2024) used the Logit Lens to explore how LLMs handle different stages of question-answering tasks. Dar et al. (2022) mapped attention weights of LLMs to lexical space, showing that these weights encode consistent concepts and relations. Belrose et al. (2023) introduced the Tuned Lens, which improves the capability and reliability of the Logit Lens. Finally, Ghandeharioun et al. (2024) proposed the Patchscopes framework, demonstrating that auxiliary models can represent lexical projections through tuning.

Mechanistic Interpretability serves as a tool for debugging and enhancing LLMs and can be applied to a variety of downstream tasks. Xiao et al. (2024) leveraged explanations from multi-head self-attention (MHSA) mechanisms in LLMs by introducing StreamingLLM, a model capable of handling unlimited text without requiring fine-tuning. Through causal tracing, Hendel et al. (2023); Todd et al. (2024) demonstrated that certain attention heads can efficiently encode compact representations of example tasks, leading to improved performance in few-shot prompting. Liu et al. (2024) explored the role of social bias in LLMs, introducing the concept of social bias neurons to explain and mitigate such biases. Furthermore, Li et al. (2024b) proposed an intervention technique during inference, which, based on the interpretability of attention heads, shifts activation values toward "truthful" responses to reduce model hallucinations. In this paper, we analyze the MLP and MHSA components of LLMs to uncover the mechanisms that enable multi-hop reasoning and, building on our findings, we introduce a targeted knowledge-editing method, IFMET.

# B. Dataset

## B.1. Details of MQuAKE-CF-3K

The MQuAKE-3K dataset comprises over 3,000 N-hop questions (where N $\in \{2, 3, 4\}$), each associated with one or more edits. We use this dataset as a diagnostic tool to evaluate the performance of edited models in integrating newly injected knowledge through editing methods. Two evaluation scenarios can be considered: a) In the first scenario, we perform knowledge editing on a single instance $d$, which may involve up to four fact edits. b) In the second scenario, the dataset is divided into groups of $k$ instances ($k \in \{1, 100, 1000, 3000\}$ for MQuAKE-3K. In this case, all instances within a group are processed simultaneously, and the edited facts of these instances are injected into the model at once. This more challenging setting is particularly relevant for editing methods like MEMIT (Meng et al., 2023), which efficiently handle large volumes of edits. The main experiments in the main text focus on the first scenario, while experiments for the second scenario can be found in the appendix G.

## B.2. Subset of MQuAKE

### B.2.1. 1-HOP AND 2-HOP SUBSET FOR MECHANISM EXPLORATION

In exploring the mechanisms of fact recall for one-hop and two-hop queries, this experiment utilized cloze templates as the experimental prompts. We extracted knowledge from MQuAKE that could be answered by GPT-J-6B in a zero-shot setting. This approach ensured that the model could recall the knowledge under the strictest conditions while minimizing the impact of unclear responses on the experimental results. The distribution of various relation types across the two subsets is illustrated in Figure 4.

### B.2.2. PRE AND POST SUBSET

To construct the subset, we selected two-hop queries from MQuAKE with Cloze-Format templates and then randomly drew a nearly equal number($\approx 300$) of cases based on the proportion of relations.

# C. Causal Intervention

## C.1. Least squares and Minimum-norm method

When performing interventions, we need to solve the least squares constraint as follows:

$$\arg\min_{\Delta h_l} \|W_U(h_l + \Delta h_l) - s_l^*\|^2$$

In certain situations, the minimum norm method is more effective than directly solving linear systems or using other numerical methods, especially when the system is underdetermined (i.e., there are fewer equations than unknowns) or when there are infinitely many solutions. The minimum norm method provides a solution with the smallest norm among all possible solutions.

To minimize the probability of the intermediate answer $j$, we replace its logits with the smallest logits of the model's vocabulary and provide appropriate compensation for the final answer $k$ to maintain the probability of the final answer unchanged. The $\Delta h$ can be represented as:

$$\begin{cases} \Delta h = \Delta h_j + \Delta h_k \\ \Delta h_j = \frac{s_l[j] - s_l^{\min}}{\|W_u[j]\|^2} W_u[j] \\ \Delta h_k = \frac{s_l[k] - s_l^{\min}}{\|W_u[j]\|^2} \alpha W_u[k] \end{cases}$$

The change in the probability of the final answer after causal intervention can be represented by the function $f(\alpha)$: $f(\alpha) = P(h^*, k) - P(h, k)$ Where $f(\alpha)$ is a monotonically increasing function on the interval $(0, 1)$. We can find the zero of this function using the bisection method, ensuring that the final answer, after the causal intervention, remains within an acceptable error margin with unchanged probability.

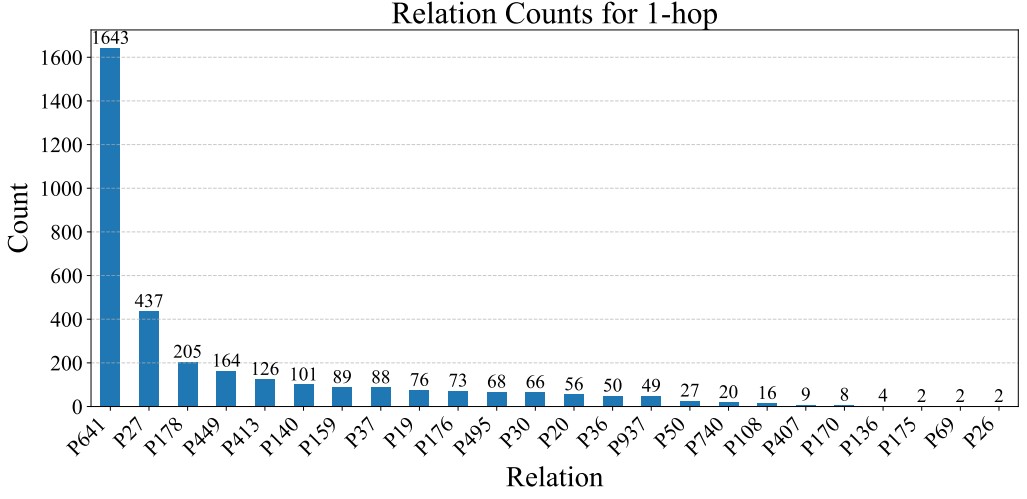

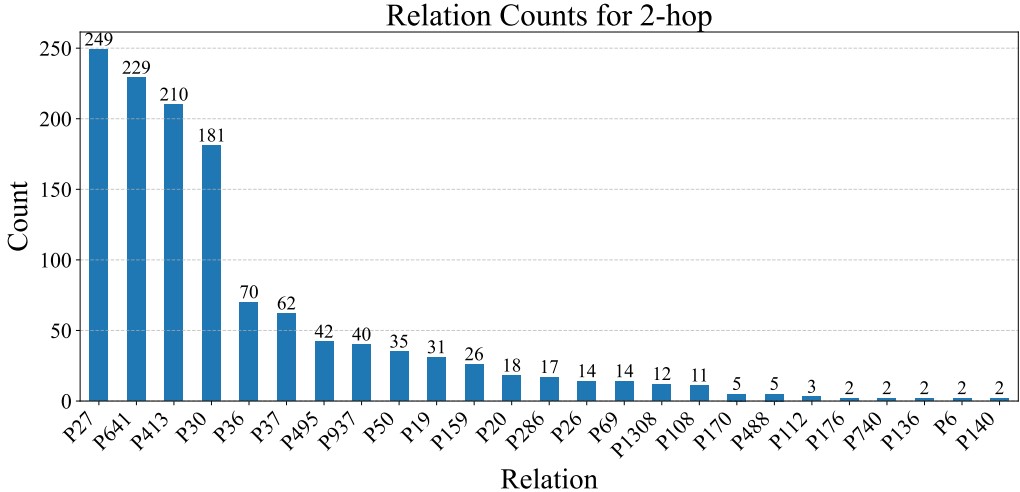

*Figure 4.* Relation number for 1-hop and 2-hop

## C.2. Comparative experiments on the same fact at different positions in multi-hop factual recall

We sampled two subsets with approximately equal sizes from the MQuAKE-3K dataset, detailed in Appendix B.2.2. By applying the SOTA locate-then-edit method PMET to layer [3-8], which follows (Li et al., 2024c), we present the percentage of cases where both pre-edited and post-edited models answer successfully in QA format or Cloze format under different edit batches. Notably we show the performance of pre-edited model on unedited answer, as well as the edited answer for the posted-edited model.

# D. Details of IFMET

## D.1. Detailed Multi-hop Edit Prompts Construction Process

We provide two methods for constructing multi-hop edit prompts. The first method is used in the main experiments of this paper, while the performance of multi-hop edit prompts generated by the second method is discussed in the appendix G.

**WikiData as Knowledge Base** Practically, we utilize WikiData[5] to construct. We start by extracting all 2615 subjects from the MQuAKE dataset's edits and deduplicating them to form a set of subjects $S_e = \{s_i | i = 1, \dots \}$. For each subject $s$, We then perform a WikiData SPARQL query[6] to identify a set of triplets for each subject $s_i$: $Sup = \{(s', r', o') | o' = s_i\}$. The query is illustrated in Table 14. To keep the query complexity within an acceptable range, we collected all relationships that have appeared in MQuAKE and restricted $r'$ to those that have occurred in the relation set. To ensure the reliability of these facts, we then use the prompt 11 to filter out the answerable $(s', r', s)$ triples. For each edit case $(s, r, o \rightarrow o*)$, we are able to construct a two-hop edit template $T_C(s')$ with the multi-hop chain $C = (s', r's) \oplus (s, r, o \rightarrow o*)$.

**Model as Knowledge Base** We used a simple prompt in table 13 to retrieve relevant knowledge directly from the model to be edited for constructing the multi-hop edit prompt. Due to computational and time constraints, we limited each case to a minimum of one multi-hop edit prompt and a maximum of five multi-hop edit prompts.

## D.2. Detailed Edit Process

---

**Algorithm 1:** IFMET

**Data:** Requested edits $E = \{(s_i, r_i, o_i \rightarrow o_i^*)\}_{i=1}^N$, Supplementary set $Sup = \{(s_i', r_i', s_i)\}_{i=1}^N$, model $\mathcal{M}$, first edit layers $l_1$, furtherance edit layers $l_2$

**Result:** Modified model $\mathcal{M}_E$ containing edits from $E$

1 **for** $\underline{(s_i, r_i, o_i^*) \in E}$ **do**              // First Edit Process
2      **Generate the single edit prompt** $T_{r_i}(s_i)$ ;
3      **Optimize** $v_i^* \leftarrow Search(T_{r_i}(s_i))$ ;         // $v_i^*$ for every new fact
4 **end**
5 **for** $\underline{l \in l_1}$ **do**                // Update weights of Shallow MLPs
6      $\Delta^l \leftarrow Calculate([v_1^*, \dots, v_N^*])$    ;       // Compute weight change with target vectors
7      $W^l \leftarrow W^l + \Delta^l$    ;            // Update layer $l$ MLP weights in model
8 **end**
9 **for** $\underline{(s_i', r_i', s_i) \in Sup}$ **do**          // Furtherance Edit Process
10      **Construct the multi-hop Chain** $C = (s_i', r_i', s_i) \oplus (s_i, r_i, o)$ ;
11      **Generate the multi-hop edit prompt** $T_C(s_i')$ ;
12      **Optimize** $v_i^* \leftarrow Search(T_C(s_i'))$ ;
13 **end**
14 **for** $\underline{l \in l_2}$ **do**              // Update weights of Deeper MLPs
15      $\Delta^l \leftarrow Calculate([v_1^*, \dots, v_N^*])$ ;
16      $W^l \leftarrow W^l + \Delta^l$
17 **end**

---

Our method primarily consists of a first edit (step 1-8 in Algorithm 1) and a furtherance edit (step 9-17 in Algorithm 1).

---

[5] www.wikidata.org
[6] https://query.wikidata.org/

Each single edit process obtains target weights via optimizing the objective of knowledge preservation and editing:

$$\underset{\hat{W}}{\arg\min} \left( \lambda \underbrace{\|\hat{W}K_0 - W^{out}K_0\|^2}_{\text{Preserve}} + \underbrace{\|\hat{W}K_E - V_E\|^2}_{\text{Edit}} \right),$$

where $K_0 = \left[ k_0^1 \,|\, k_0^2 \,|\, \cdots \,|\, k_0^N \right]$ and $V_0 = W^{out}K_0$ contain all the knowledge we want to preserve, $K_E = \left[ k_e^1 \,|\, k_e^2 \,|\, \cdots \,|\, k_e^E \right]$ is the matrix containing the edits we try to make and $V_e = [v_{e_1}^* \,|\, \ldots \,|\, v_{e_E}^*]$ represents the target representations of the new knowledge. $(K_E, V_E)$ corresponds to the edited fact set $\{(s_i, r_i, o_i^*)|i = 1, 2, \cdots, E\}$. We consider the target weight $\hat{W}$ as the sum of the original weight $W^{out}$ and the incremental weight $\Delta$, as explicated in (Li et al., 2024c), a closed-form solution to the incremental weight can be derived:

$$\Delta = RK_E^T(C_0 + K_E K_E^T)^{-1}, \quad R \triangleq (V_E - W^{out}K_E), \quad C_0 \triangleq K_0 K_0^T. \tag{3}$$

Thus, solving the optimal parameter $\hat{W}$ is transformed into calculating edited fact representation $\{(k_e^i, v_e^i)|i = 1, \ldots, E\}$. In this process, an edit instance $e = (s, r, o \to o*)$, $(k_e, v_e)$ the pre-edited fact $(s, r, o)$ and $(k_e, v_e^*)$ denotes post-edited $(s, r, o*)$. To obtain the target representations of the new knowledge $v_e^* = v_e + \delta$, we optimize the learnable parameter vector $\delta$ to modify the original value vector. $Search$ is the process of obtain the optimized $\delta$ through gradient descent:

$$\delta = \underset{\delta}{\arg\min} \mathcal{L}(\delta) = \mu D_{\text{KL}} \left( P_{\mathcal{M}_e} [t' \mid T] \,\|\, P_{\mathcal{M}} [t' \mid T] \right) + \varphi \frac{1}{P} \sum_{j=1}^{P} - \log \mathbb{P}_{\mathcal{M}_e} \left[ o^* \mid \text{pref}_j \oplus T_e \right],$$

where $T$ is the KL prompt, such as "$s$ is a " and $t'$ is the tokens excluding the token for the answer $o^*$, $T_e$ is the prompt for editing, such as "*The capital of Spain is* ", $\varphi$ and $\mu$ serve as the scaling factor for adjusting the loss. $Calculate$ process is using the $v_e^*$ to slove the $\Delta$ which is a function of $v_e^*$. involves substituting the values of $V_e = [v_{e_1}^* \,|\, \ldots \,|\, v_{e_E}^*]$ corresponding to a series of edits into (3) to compute the $\Delta$.

The primary differences between the first and furtherance edits are reflected in the edit prompt $T_e$ and the layers edited. For example, for the edit instance $e = (s, r, o \to o*)$, the first edit utilized a single-hop edit template $T_e = T_r(s)$ provided by MQuAKE to edit layers [3,8] of the GPT-J model in the subject last token position. For the furtherance edit, a two-hop prompt $T_e = T_C(s')$ composed of a support case $(s', r, s)$ and $(s, r, o^*)$, and this two-hop edit prompt was applied to edit layers [16,20] of the GPT-J model in the last token position.

# E. Addition Experimental Settings

### E.1. Criteria for classifying dataset into *pre* and *post*.

Consider a multi-hop question composed of $n$ triples. We define the positions of edits (with the index starting from 1) as the set $\{e_1, e_2, \ldots, e_m\}$, where $m$ represents the total number of edits. The important insight is to divide according to whether only the first hop is edited. Edits only occurring in the first hop are classified as *pre*, while those hold edits more than one are labeled *post*.

### E.2. Experimental Settings

When constructing the multi-hop edit prompts, for each edit case, no more than three triplets per relation were added. The relation types of the multi-hop edit prompts set are the same as MQuAKE. We set the edit batch sizes to 1.

In both the first and furtherance edits, our configuration for PMET adheres to the settings specified by (Li et al., 2024c). Initially, we set $\varphi = 1$ and $0 \leq \mu \leq 1$ to manage the retention of the model's original knowledge. As $\mu$ increases, the retention level also increases, while $\varphi$ exhibits the opposite trend. After maximizing the probability of the target knowledge, we reduce $\varphi$ to 0.1 to preserve the original knowledge as much as possible. Optimization is halted when $D_{\text{KL}} < 0.01$. On GPT-J, for estimating the covariance matrix (i.e., the set of previously memorized keys $C_0$), we sample 10,0000 times on Wikitext in fp32 precision and set $\lambda = 6000$. When optimizing, we limit the total optimization steps to 30 with a learning rate of 0.2. All our experiments were conducted using the MQuAKE dataset. To test the accuracy of answers to multi-hop questions, we adhered to the few-shot in Table 12 and Chain of Thought (CoT) templates in Table 10 and procedures as outlined in (Zhong et al., 2023).

# F. Ablation Study

Our mechanism analysis has identified that the existing editing methods with only single-hop edit prompts fail to adequately modify knowledge in the deeper MLP layers, resulting in poor performance on multi-hop factual recall tasks. Additionally, our findings suggest that implicit multi-hop step dependencies rely on the knowledge provided by these deeper MLP layers. Based on these interpretability results related to the mechanistic difference, we propose the **IFMET**. Given the distinctions between **IFMET** and other existing methods, we highlight four key components, especially in the furtherance edit: two-stage modification, the use of multi-hop edit prompt, editing the last token position, and updating knowledge in the deeper-layer MLPs during the second stage, as illustrated in the table 6.

To investigate whether the IFMET method effectively balances the requirements of general knowledge editing and multi-hop fact recall tasks, we constructed the paraphrase set and neighborhood set for a subset of the MQuAKE-3K dataset , following the approach used in the COUNTERFACT dataset (Meng et al., 2022b). We conducted experiments under two configurations: edit batch = 1 and edit batch = 100 and focused on analyzing the roles of these key components on the following metrics:

- **Efficacy** measures whether an edit has been successfully applied to a model and whether the model can successfully answer the single-hop fact recall prompt. It is calculated as the percentage of edits where $P(\text{edited answer}) > P(\text{unedited answer})$ for a given query prompt used during model editing.

- **Multi-hop** measures whether the multi-hop question related to the edited knowledge can be successfully answered by the model.

- **Paraphrase** evaluates the model's generalization ability under an edit. It is defined as the percentage of edits where $P(\text{edited answer}) > P(\text{unedited answer})$ for paraphrases of the query prompt.

- **Specificity** assesses the locality of the model editing, i.e., whether the edit of a specific fact affects other facts stored within the model. Specificity score is defined as the percentage of facts in the neighborhood of the edited fact that remain unchanged after the edit.

In Table 7, we have utilized the **PMET** as a baseline to assess method performance. The importance of each component is reflected through comparisons of performance improvements over **PMET**. **PMET**'s performance exemplifies a single-stage locate-then-edit approach editing shallow MLPs based on a single-hop prompt. From the analysis of the ablation experiments, we derive the following conclusions:

- **IFMET**: Firstly, it can be observed that the implementation of **IFMET** achieves the best performance in Multi-hop Acc. In the second stage of editing, we employ multi-hop edit prompts alongside deep MLP editing techniques. Across all the experimental tables mentioned, **IFMET** consistently demonstrates a substantial improvement in inferential performance compared to **PMET**. As the edit batch size increases, IFMET continues to sustain performance improvements. However, it is worth noting that the IFMET framework also leads to a slight decrease in the specificity metric, indicating that more extensive modifications to the model's knowledge can somewhat compromise specificity. But, as mentioned in previous works, the pursuit of a balanced performance across the four metrics—Multi-hop Efficacy, Specificity, Paraphrase—we believe IFMET performs exceptionally well in this regard as well.

- **w/o** $First$: When only applying the second edit stage in IFMET, which modifies the deeper layers using a multi-hop edit prompt, it effectively enhances performance on multi-hop reasoning tasks. However, the absence of first-stage editing results in unchanged knowledge in the earlier layers, leading to poor performance in single-hop fact recall tasks. This highlights the importance of the two stages, where each stage is indispensable.

| Method | Stage | Edit Prompt | Position | Layers |
|--------|-------|-------------|----------|--------|
| Previous | Only one | Single-hop | Subject Last Token | Shallow |
| IFMET | First Furtherance | Single-hop Multi-hop | Subject Last Token Last Token | Shallow Deeper |

*Table 6.* The main difference between **IFMET** and previous methods. The term **Stage** refers to the phases of the editing process, **Data** denotes the query utilized for editing, **Position** specifies the token position where the editing is applied, and **Layers** indicate the edited layers.

| Edits | Editor | Multi-hop | Efficacy | Specificity | Paraphrase |
|-------|--------|-----------|----------|-------------|------------|
| | IFMET | 28.38 (↑**78.0%**) | 99.56 (↑12.8%) | 69.87 (↓11.1%) | 90.17 (↑**5.3%**) |
| | w/o $First$ | 23.14 (↑45.1%) | 66.59 (↓**24.6%**) | 62.61 (↓**20.3%**) | 41.48 (↓**51.6%**) |
| | w/o $Multi$ | 17.69 (↑10.9%) | 100.00 (↑**13.3%**) | 77.71 (↓1.2%) | 86.24 (↑0.7%) |
| 1 | w/o $Last$ | 18.12 (↑13.6%) | 88.21 (↑0.0%) | 78.60 (↓**0.0%**) | 86.24 (↑0.7%) |
| | w/o $Deeper$ | 15.07 (↓**5.4%**) | 99.56 (↑12.8%) | 70.31 (↓10.6%) | 86.90 (↑1.5%) |
| | PMET | 15.94 | 88.21 | 78.60 | 85.59 |
| | IFMET | 27.07 (↑**64.8%**) | 96.29 (↑8.1%) | 69.89 (↓9.4%) | 84.28 (↑**3.8%**) |
| | w/o $First$ | 22.71 (↑35.1%) | 73.36 (↓**17.8%**) | 69.21 (↓**10.2%**) | 34.72 (↓**57.3%**) |
| | w/o $Multi$ | 17.25 (↑2.6%) | 99.13 (↑11.3%) | 76.63 (↑**0.6%**) | 84.06 (↑3.5%) |
| 100 | w/o $Last$ | 15.94 (↓**5.2%**) | 89.08 (↓0.0%) | 76.85 (↑0.3%) | 81.55 (↑0.4%) |
| | w/o $Deeper$ | 16.16 (↓3.9%) | 99.56 (↑**11.8%**) | 74.67 (↓3.2%) | 81.00 (↓0.3%) |
| | PMET | 16.81 | 89.08 | 77.07 | 81.22 |

*Table 7.* The results of the ablation experiments on GPT-J-6B model using a subset of MQuAKE-3K. Both the percentages of decrease(↓) and increase(↑) are calculated relative to **PMET** as the baseline. The most significant performance decline is highlighted in **red** and the most significant performance increase is highlighted in **green**.

- **w/o** $Last$ demonstrated the importance of editing the last token position in the second stage. This is consistent with the findings from the mechanism exploration in Section 3.1, where subject information converges at the last token position and has a causal effect on the retrieval of the knowledge about the final answer.

- **w/o** $Multi$: This represents that, in the second editing stage, we continued to use the original single-hop edit prompt instead of the multi-hop edit prompt to edit the deeper MLP layers. The performance improvement in efficiency somewhat indicates that single-layer edit prompts can edit knowledge in deep MLP for single-hop scenarios, but there is no significant improvement in multi-hop performance. This suggests that, due to the difference in mechanisms, the knowledge used in implicit reasoning steps that have not been sufficiently modified in the deep MLP layers remains unedited. It also indicates that using single-hop prompts combined with deep MLP editing is ineffective, highlighting the critical importance of the multi-hop prompts.

- **w/o** $Deeper$: In this setup, the second-stage editing was modified to use the multi-hop edit prompt combined with shallow MLP editing(rather than deeper MLP layers). If this also shows a significant performance improvement in multi-hop accuracy, it would indicate that merely expanding with the multi-hop edit prompt, without considering its mechanism on the deeper MLP layers, can enhance results. However, as observed across the table, there was a consistent minor fluctuation in performance (ranging from -5.4% to +3.9%). In contrast to **IFMET**'s own +70% improvement, this underscores the importance of editing the deeper MLP layers when using the multi-hop edit prompts set.

In light of the results from the ablation experiments **w/o** $Multi$ and **w/o** $Deeper$, we emphasize that it is essential to apply multi-hop edit prompts to the deeper MLP layers for knowledge editing, ensuring a mechanism match, in order to effectively improve the performance of multi-hop factual recall tasks. In conclusion, we can summarize that the four components proposed by IFMET are essential for performance improvement, also highlighting the importance and validity of our interpretability exploration work.

## G. Generalizability of IFMET

In this subsection, We explore the generalizability of our method from four key perspectives:

**Generalization to other models.** We repeated the ablation experiments on LLaMA-2-7B to evaluate the generalizability of **IFMET**. The results shown in Table 8 are consistent with those observed on GPT-J, highlighting the importance of the four components in **IFMET** as well as the superiority of the method itself on LLaMA-2-7B model. Considering both the interpretability analysis and experimental outcomes, we conclude that our analysis and method are equally applicable to larger models, such as LLaMA-2.

**Generalization to other edit batch number.** In addition to editing a single instance at a time, we also tested scenarios

| Edits | Editor | Multi-hop | Efficacy | Specificity | Paraphrase |
|-------|--------|-----------|----------|-------------|------------|
| | IFMET | 28.38 (↑**73.3%**) | 99.78 (↑**13.7%**) | 65.50 (↓10.8%) | 75.00 (↑**23.1%**) |
| | w/o $First$ | 25.76 (↑57.3%) | 56.55 (↓**35.6%**) | 62.18 (↓**15.3%**) | 39.08 (↓**35.9%**) |
| | w/o $Multi$ | 19.43 (↑18.6%) | 98.68 (↑12.4%) | 70.35 (↓4.2%) | 69.00 (↑13.2%) |
| 1 | w/o $Last$ | 15.72 (↓4.1%) | 88.86 (↑1.2%) | 73.03 (↓**0.6%**) | 61.90 (↑1.6%) |
| | w/o $Deeper$ | 15.28 (↓**6.8%**) | 96.72 (↑10.1%) | 65.61 (↓10.7%) | 66.99 (↑9.9%) |
| | w $Multi_{model}$ | 26.86 (↑64.0%) | 96.07 (↑9.4%) | 63.32 (↓13.8%) | 74.24 (↑21.8%) |
| | PMET | 16.38 | 87.77 | 73.41 | 60.92 |
| | IFMET | 27.29 (↑**76.0%**) | 97.82 (↑4.2%) | 65.50 (↓9.9%) | 84.17 (↑**14.2%**) |
| | w/o $First$ | 24.67 (↑59.2%) | 64.41 (↓**31.4%**) | 63.97 (↓12.0%) | 41.81 (↓**43.3%**) |
| | w/o $Multi$ | 15.07 (↓2.8%) | 99.34 (↑**5.8%**) | 71.83 (↓1.2%) | 76.64 (↑4.0%) |
| 100 | w/o $Last$ | 13.97 (↓**9.9%**) | 94.32 (↑0.4%) | 72.14 (↓**0.8%**) | 74.67 (↑1.3%) |
| | w/o $Deeper$ | 15.94 (↑2.8%) | 96.51 (↑2.8%) | 69.48 (↓4.4%) | 75.44 (↑2.3%) |
| | w $Multi_{model}$ | 22.49 (↑45.1%) | 96.07 (↑2.3%) | 63.32 (↓**12.9%**) | 79.26 (↑7.5%) |
| | PMET | 15.50 | 93.89 | 72.66 | 73.69 |

*Table 8.* The results of the ablation experiments on LLaMA-2-7B model using a subset of MQuAKE-3K. **w** $Multi_{model}$ represents the multi-hop edit prompts generated by model itself to modify deeper MLPs. Both the percentages of decrease(↓) and increase(↑) are calculated relative to **IFMET** as the baseline.

where 1,000 and 3,000 instances are edited together. The results, shown in Figure 5, demonstrate that IFMET outperforms existing methods across all edit batch sizes.

**Construction of the multi-hop prompts.** In **IFMET**, we emphasize the importance of constructing a multi-hop edit prompt for each edit instance. It is important to note that any other valid knowledge base can replace WikiData and SPARQL. A straightforward alternative is to treat the model itself as a reliable knowledge base for extracting relevant knowledge.

The results of substituting the multi-hop edit prompt with one generated by LLaMA-2 itself for editing are also shown in Table 8 called **w** $Multi_{model}$. The results show a significant improvement over the one-stage **PMET**, with performance trends aligning closely with those of **IFMET**. Notably, minimal effort was invested in designing the knowledge retrieval prompt, and no additional filtering or preprocessing was applied. This suggests that the multi-hop edit prompts generated by the model represent a relatively low-quality version, effectively serving as a lower bound for the method's performance across various metrics. Despite this, it still outperforms existing one-stage methods. This highlights the inherent superiority of the **IFMET** framework and demonstrates the feasibility of using the model itself to construct the prompts set.

**Time complexity of IFMET.** We compared the time complexity of **IFMET** with that of the one-stage **PMET** method it

| Model | Method | Time |
|-------|--------|------|
| | MEMIT | 4.5s |
| GPT-J-6B | PMET | 5.0s |
| | IFMET | 9.7s |
| | MEMIT | 2.1s |
| LLaMA-2-7B | PMET | 2.0s |
| | IFMET | 3.4s |

*Table 9.* The average time required to edit a single case varies across methods. For the two one-stage methods, **MEMIT** and **PMET**, this corresponds to the process of optimizing the shallow-layer MLPs using single-hop queries. For **IFMET**, the process includes updating the deeper-layer MLPs using two-hop edit prompts.

builds upon, the result is shown in Table 9. On average, the time required to perform a complete edit for a single case on GPT-J using **IFMET** was approximately 2.5× that of **PMET**. For LLaMA-2, the time required was about 1.5× that of **PMET**. We believe this is within an acceptable range, and as the editing speed of the single-stage method improves, the **IFMET** framework will correspondingly become faster.

### G.1. Comparison with Weight-Preserving Methods

Although there have been some weight-preserving editing methods(e.g. RAG-based Methods) accessing good performance for multi-hop question answering in the KE scenario, we believe that exploring the locate-then-edit methodology remains meaningful for several reasons:

1. **From the perspective of understanding internal knowledge utilization**: The mechanisms underlying a model's use of internal knowledge differ fundamentally from those governing the use of external knowledge (Jin et al., 2024). Investigating the potential of locate-then-edit methods holds significant value for advancing the interpretability of internal knowledge processes, laying the groundwork for deeper insights and practical implementations. Additionally, we believe this approach enables a more fundamental and precise modification of knowledge.

2. **From a practical standpoint**: Methods based on retrieval-augmented generation (RAG) require providing extensive contextual input tokens, posing substantial challenges in terms of computational efficiency and hardware demands. And these methods face several challenges. Instead of injecting knowledge into LLMs, they retrieve related facts stored in memory for editing. As a result, their retrieval success rates become crucial, particularly when managing complex real-world scenarios involving exponential growth in knowledge updates. Moreover, we argue that an over-reliance on modifying knowledge through external contexts introduces security risks, as it may be exploited for data theft and attacks (Upadhayay et al., 2024), especially in real-world applications.

## H. Additional Experimental Results

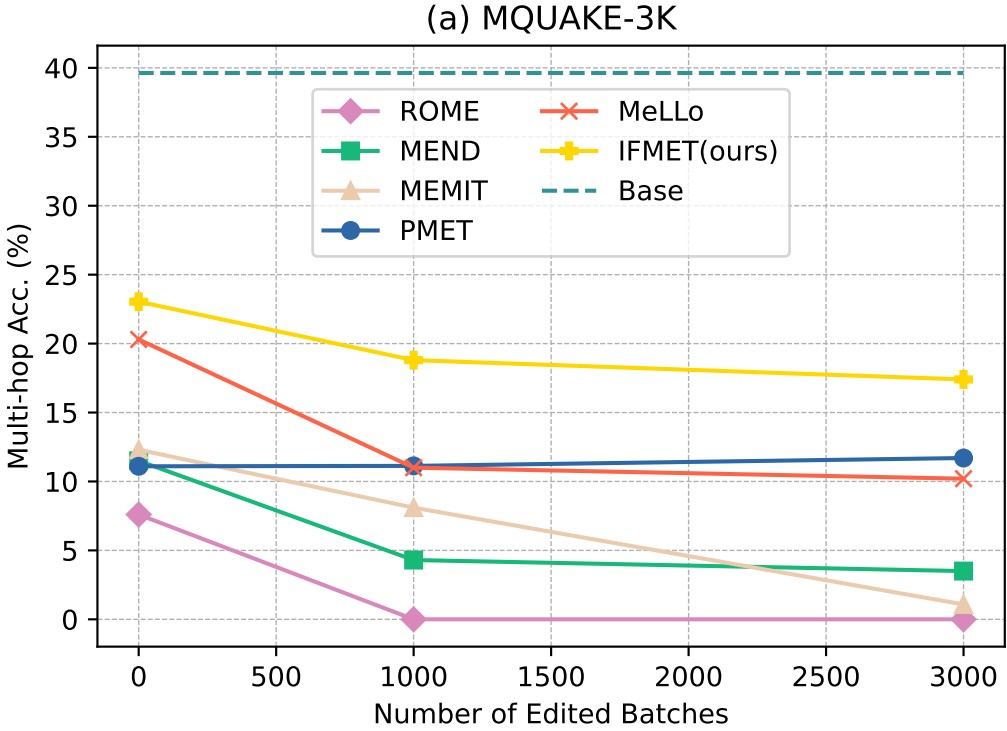

*Figure 5.* **Multi-hop Acc** Performance Comparison of different methods across edit batch sizes on MQUAKE-3K. **Base** in this table represents unmodified GPT-J-6B model, and we report its performance on **unedited answer**.

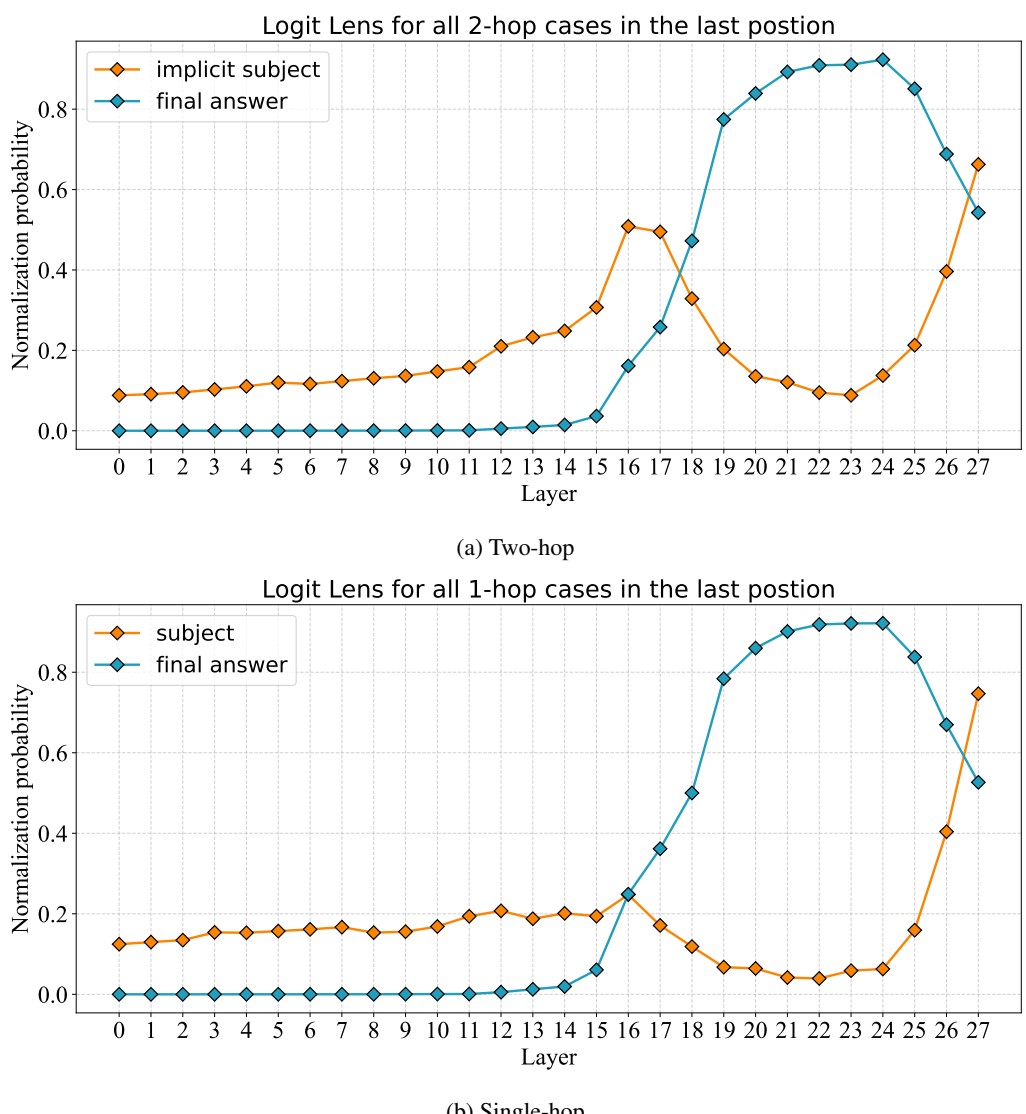

(a) Two-hop

(b) Single-hop

*Figure 6.* **LogitLens results of the last token position at different layers.** (a) Yellow line represents the information containing implicit subject $s_2$, i.e., Info$(h_l, s_2)$. Blue line represents the information for the final answer, i.e., Info$(h_l, o_2)$. (b) Yellow line represents the information of subject $s$, i.e., Info$(h_l, s)$, and Blue line represents the information of the answer $o$, i.e., Info$(h_l, o)$. Larger versions of the sub-figures are available in the Appendix.

# I. Prompts And Templates

```
Question:  What is the capital of the country where Plainfield Town Hall is located?
Thoughts:  Plainfield Town Hall is located in the country of the United States of America.
The capital of United States is Washington, D.C.
Answer:  Washington, D.C.

Question:  In which country is the company that created Nissan 200SX located?
Thoughts:  Nissan 200SX was created by Nissan.  Nissan is located in the country of Japan.
Answer:  Japan

[3 in-context demonstrations abbreviated]

Question:  Who has ownership of the developer of the Chevrolet Corvette (C4)?
Thoughts:  The developer of Chevrolet Corvette (C4) is Chevrolet.  Chevrolet is owned by
General Motors.
Answer:  Model Generated Answer Goes Here
```

*Table 10.* The template of the prompt we used for asking multi-hop questions using chain-of-thoughts. Our CoT setting is to cut the prompt after **Thoughts:**, so that the LM has to generate the thought itself (pressured by the prompt and ICL examples). For the same multi-hop problem, we generate cot and its corresponding answers three times to ensure that possible errors are avoided.

```
(In-context-learning examples)
Q: Who is the developer of Telegram?  A: Telegram FZ-LLC
Q: Who is the developer of Microsoft Windows?  A: Microsoft
Q: Who is the developer of PlayStation 2?  A: Sony Interactive Entertainment
Q: Who is the developer of iTunes?  A: Apple Inc.
Q: Who is the developer of SR-71 Blackbird?  A: Kelly Johnson
Q: Who is the developer of Moblin?  A: Linux Foundation
Q: Who is the developer of Xbox 360?  A: Microsoft
Q: Who is the developer of Kinsey scale?  A: Alfred Kinsey
(Query during inference)
Q: Who is the developer of SteamOS? A:Valve Corporation
```

*Table 11.* An example of the prompt we used to recall single-hop fact

```
(In-context-learning examples)
Q: What is the country where The Rotunda is located?  A: United States of America
Q: In which country was Tohar Butbul granted citizenship?  A: Israel
Q: Who was Nissan 200SX created by?  A: Nissan
Q: What continent is the country where Prickly Pear grows located in?  A: Europe
Q: What is the capital of the country where Plainfield Town Hall is located?  A:
Washington, D.C.
Q: In which country is the company that created Nissan 200SX located?  A: Japan
Q: Who was Dodge Ram SRT-10 created by?  Dodge
Q: Who is the spouse of Joe Biden?  A: Jill Biden
Q: Which continent is the country where the director of "My House Husband:  Ikaw Na!" was
educated located in?  A: Asia
Q: What country was the location of the Battle of Pressburg?  A: Hungary
Q: Who is the spouse of the US president?  A: Jill Biden
Q: Who has ownership of the developer of the Chevrolet Corvette (C4)?  A: General Motors
Q: Who is Joe Biden married to?  A: Jill Biden
Q: What is the country of citizenship of Charles II of Spain?  A: Spain
Q: Who was Chevrolet Biscayne created by?  A: Chevrolet
Q: What is the name of the current head of state in United Kingdom?  A: Elizabeth II
Q: multi-hop question
```

*Table 12.* The template of the prompt we used for asking multi-hop questions using few shot.

```
(In-context-learning examples)
Input:  The country that has nationals <mask> is located in the continent of Asia
Output:  Hitomi Yaida
Input:  The country that has nationals <mask> has the official language of Italian
Output:  Giorgio Chiellini
Input:  The university where <mask> was educated located its headquarters in the city of
Vienna
Output:  Michael Haneke
Input:  The country that has nationals <mask>, its capital is Washington
Output:  Lou Pearlman
Input:  The person who found <mask> is a citizen of United States of America
Outout:  Microsoft
Input:  The creator of <mask> hails from Italy
Output:  Ferrari
Input:  The author of <mask> is a citizen of United States of America
Output:  Holly Potter
Input:  The person who discovered <mask> lives in Germany
Output:  Volkswagen
Input:  question
```

*Table 13.* The template of the prompt we used for asking LLaMA-2-7B to generate the multi-hop edit prompt

```
SELECT ?subject ?subjectLabel ?predicate ?predicateLabel WHERE
?subject ?predicate wd:ss.
FILTER (?predicate IN (wdt:relation))
SERVICE wikibase:label  bd:serviceParam wikibase:language "en".
LIMIT 50
```

*Table 14.* The template of the SPARQL Query we used for prefix fact recall triplets for a specific subject.

