# OpenReview forum: "Locate-then-edit for Multi-hop Factual Recall under Knowledge Editing"
_ICML.cc/2025/Conference — ICML 2025 poster_

### Official Review · Reviewer_ixiZ · 2025-03-04

**Overall Recommendation:** 3

**Summary:**

This paper extends the *locate-then-edit* approach of model editing proposed by [Meng et al, 2023](https://arxiv.org/pdf/2202.05262) to multi-hop factual recall tasks. During the localization experiments the authors find that: on multi-hop factual recall tasks, the LM retrieves implicit subject information in the *deeper* MLP layers; which the paper claims as the key reason why existing knowledge editing methods targeting shallower MLP layers do not generalize for multi-hop factual reasoning tasks. With this finding, the authors propose IFMET, a new 2-step knowledge editing approach that edits shallower MLP layers first for single-hop fact recall tasks, and then edits deeper MLP layers for multi-hop factual recall tasks.

## update after rebuttal
I thank the authors for their detailed response to my questions. And I would like to keep my score as is.

**Claims And Evidence:**

I have several concerns on the localization experiments (Section 3.1) and evaluation of IFMET. Please see my questions.

**Essential References Not Discussed:**

Two relevant papers comes to my mind.

1. [Geva et al., 2023](https://arxiv.org/abs/2304.14767) -- investigates single-hop factual recall mechanism in LMs.
2. [Merullo et al., 2023](https://arxiv.org/abs/2305.16130) -- also investigates mostly single-hop factual recall. But the paper finds that sometimes the subject entity is being recalled again at the last token position of the prompt.

Just to be clear, I don't feel strongly about either of these papers. I thought that they are relevant to the paper and would leave it to the authors to decide whether to cite them.

**Experimental Designs Or Analyses:**

I think using Logit Lens as a tool to identify key locations/states in the LM computation has a systematic bias towards the later layers. See questions for details.

**Methods And Evaluation Criteria:**

The authors use MQuaAKE [Zhong et al, 2024](https://arxiv.org/pdf/2305.14795), which is a recognized benchmark for multi-hop factual recall tasks, to evaluate their method. However I have some concerns about the evaluation methodology. See questions.

**Other Comments Or Suggestions:**

N/A

**Other Strengths And Weaknesses:**

**Strenghts:** Well-motivated paper -- addresses a key problem in knowledge editing in LMs.
**Weaknesses:** See questions.

**Questions For Authors:**

1. **Localization of key states and modules**

    a. Section 3.1 extensively uses Logit Lens ([Nostalgebraist, 2020](https://www.lesswrong.com/posts/AcKRB8wDpdaN6v6ru/interpreting-gpt-the-logit-lens)) as a mechanistic interpretability tool to identify key locations/states in the LM computation during a multi-hop factual recall task. However, Logit Lens can give noisy results in earlier layers ([Belrose et al., 2023](https://arxiv.org/pdf/2303.08112)) and only starts giving meaningful results in deeper layers ([Geva et al., 2024](https://arxiv.org/pdf/2401.06102)). Therefore, in my opinion, using Logit Lens as a localization tool in this case has a systematic bias towards deeper layers. I would love to hear the authors' thoughts on this.

    b. I am not sure that I clearly understand the motivation behind the choice of Equations 1 and 2.
    * In Equation 1, all $j \in s_2$ are being optimized for. However, $W_U h$ is a linear operation. Thereby shouldn't the (probably weighted) mean of the rows of $W_U$ corresponding to $j \in s_2$ be sufficient? Do we really need a SGD optimization here?
    * Similarly in Equation 2, all $j \in o_2$ are being considered. However, these LMs are autoregressive and will not generate all $j \in o_2$ at once. Therefore, shouldn't we only consider the first token in $o_2$? I understand that the first token is not always enough to identify the correct generation, but it should be possible to curate a set of candidates where this is not the case, which I think is enough for the purposes of this localization experiment.

    c. I am not really sure I understand the results of Figure 3. Probably having the prompts on which the intervention is being performed would help. Here's my guess about the setup, please correct me if I am wrong.

    * The multi-hop prompt is something like `"The capital of the country where {} is located is"`. `{}` is replaced with $s_2 = \texttt{the Louvre}$ and the answer is $o_2 = \texttt{Paris}$. Then $s_l^*$ is optimized for a different $s_2 = \texttt{the Statue of Liberty}$ and the intervention is being performed on corresponding states at the last token position of the prompt (?)

    * What does "invent" label mean in Figure 3? Is it a typo for "intervention"? Also the probability *decrease* is being measured. Assuming the setup I understand is correct, is it measuring the probability decrease for `Paris` or increase for `Washington`?

    * The values in Figure 3 aren't very large, strongest effects around 0.025 for residual states and $7 \times 10^{-6}$ for MLPs. Is this significant enough to draw any strong conclusions?


2. **Evaluation**

    a. The paper mentions that for IFMET the multi-hop prompt is constructed using the WikiData and the LM itself (Lines 309-10). How much do they differ from the prompts in MQuaAKE, on which the edit is being evaluated?

    b. As the single edit performance of other methods reported in Table 3 is so low, I am assuming they are all being evaluated on multi-hop prompts. Is this correct? Then shouldn't this be reported as a grid shaped #Edits $\times$ #hops matrix instead?

    c. Looking at Table 7 in the Appendix, I wonder why the single edit efficacy of PMET below 90\% despite it being a direct optimization approach? I also encourage the authors to also include ROME for single edits and MEMIT for mutiple edits in this comparison.


3. When you incorporage CoT, did you make sure to not give the answer away? The example in Table 10 has the answer in the thought process.

    One way you can easily do that is cut-off the prompt before the answer. In your example it would be `"... Chevrolet is owned by"`, But then I think the single hop editing should also be effective here. However, probably that is not what you do as you report poor performance of single hop editing in Tables 3 and 4.

    Alternatively, you can cut the prompt after `Thoughts:`, so that the LM has to generate the thought itself (pressured by the prompt and ICL examples).

    The setup wasn't clarified in the paper (if I didn't miss it) and I think it's an important thing to clarify.

**Relation To Broader Scientific Literature:**

Previous works investigating factual recall mechanisms on LMs attributed knowledge retrieval (also known as enrichment, detokenization, ...) to MLPs is shallower layers, at the subject last token position ([Meng et al, 2023](https://arxiv.org/pdf/2202.05262), [Geva et al., 2023](https://arxiv.org/abs/2304.14767)). This paper extends the setting to multi-hop factual recall tasks, and finds that the LM retrieves implicit subject information in the *deeper* MLP layers, at the last token position. The paper further shows that a knowledge editing method designed with this insight is more effective in multi-hop factual recall than previous methods targeting shallower MLP layers.

**Theoretical Claims:**

The paper is empirical in nature, and does not contain any strong theoretical claims.

---

> ### Author Rebuttal · Authors · 2025-04-01
>
> Thank you very much for your recognition of our work. Here is our response.
>
> > Response to Q1.a in Questions For Authors
>
> In the past exploration process, we also used patchscope to perform the experimental effects shown in figure 2, and judged whether the corresponding information was contained by counting whether the subsequent output of patchscope contained the answer. During the experiment, the results were surprisingly consistent with logitlens, so considering the universality of the logitlens method and cost-effectiveness, we still use it as our interpretability tool. We believe that the results obtained by logitlens are already trustworthy, but in the future, we can continue to explore further through methods such as tuned-lens.
>
> > Response to Q1.b in Questions For Authors
>
> First,regarding the optimization of s2 related information, we did not use SGD optimization, but replaced the logits of the token in s2 with the minimum value on the whole vocabulary. Here it should be expressed as $s_l^*[j]$ = min($s_l$) in equations 1, where $s_l$ represents the all tokens in vocabulary, and then a simple mathematical calculation is performed to get the corresponding hidden state with a combination of least squares and minimum-norm methods.
> And, we are sorry that we omitted explicit description of the Equation 2 in the paper. Your understanding is completely correct. We did not take all j∈o2 tokens into account. In practice, we only consider the first token in o2. As you said, this is enough for the purpose of this localization experiment.
>
> > Response to Q1.c in Questions For Authors
>
> Corresponding to the example you gave, it is a two-hop fact recall chain. It includes (the louvre, located country, france) + (France, capital, Pairs). Instead of using a counterfactual example to perform a conventional corruption, we refer to [1] and reduce the amount of information about **France** in the last token position by minimizing its corresponding logits value. Then, we observe the causal effect of **France, the implicit intermediate subject,** by observing the decrease in the probability of **Paris**.
> And the "invent" is a spelling error, and we will correct it in the camera-ready version.
>
> Regarding the issue you mentioned about the significance of causal effects, we have provided a detailed explanation in the section "**Response to Q1.2 & Q1.3 & Q1.4 in Claims And Evidence about the causal intervention experiments**" in our reply to **Reviewer JTzm**. Please refer to that section for more details.
>
> [1] Understanding and Patching Compositional Reasoning in LLMs
>
> > Response to Q2 in Questions For Authors
>
> We construct a multi-hop prompt to put the knowledge to be modified into the implicit reasoning step. While mquake only provides a single edit template, we give a practical example here.
>
>
> | Edit Case | Single-hop Prompt in MQuAKE | Test multi-hop Prompt in MQuAKE | Our multi-hop Prompt |
> | -------- | -------- | -------- | -------- |
> | (Marc Cherry,cizizen,United States of America ->  Bulgaria)  | Marc Cherry is a citizen of | Which country is the creator of \"Devious Maids\" a citizen of?     | The creator of Desperate Housewives is a citizen of    |
>
> We use all the relationships that have appeared in MQuAKE to construct possible multi-hop edit prompts and then delete those that have the same explicit subject as the test multi-hop question, keeping the best one that the model can successfully answer.
>
> In Table 3, the other methods we reported only used single-hop editing prompt, and only IFMET used our framework. For the application of multi-hop prompt in other methods, please refer to the section in the reviewer 1ooJ's response. We reported in detail the improvement achieved by applying our framework to **MEMIT**, which illustrates the generalization of our method for editing baselines.
>
> > Response to Q3 in Questions For Authors about the CoT.
>
> We are sorry that our examples are confusing to you. We may not have explained them clearly enough. As you said, our cot setting is to cut the prompt after `Thoughts:`, so that the LM has to generate the thought itself (pressured by the prompt and ICL examples). For the same multi-hop problem, we generate cot and its corresponding answers three times to ensure that possible errors are avoided. We will correct this in the new version PDF.

---

> > ### Comment · Reviewer_ixiZ · 2025-04-01
> >
> > I appreciate the authors' response to my questions. However, I would like to keep my score to 3.
> >
> > Congratulations to the authors for their work and the overall positive feedback from the reviewers. I look forward to seeing the final version of the paper in the conference.

---

> > > ### Author Response · Authors · 2025-04-05
> > >
> > > Thank you very much for the time and effort you dedicated during the review process, which significantly improved the quality of our manuscript. We also appreciate your recognition of our work and will incorporate all the valuable insights provided during the rebuttal phase into the final version

---

### Official Review · Reviewer_1ooJ · 2025-03-08

**Overall Recommendation:** 5

**Summary:**

This paper investigates how LLMs handle multi-hop factual recall under knowledge editing. Using various interpretability methods, the authors uncover a critical insight: for multi-hop questions, LLMs rely primarily on implicit subject information encoded in deeper MLP layers to derive final answers. This mechanism differs significantly from that of single-hop tasks, explaining the failure of previous locate-then-edit methods when applied to multi-hop factual recall. Building on these findings, they introduce IFMET, a novel locate-then-edit knowledge editing method. Besides editing the shallow MLP layers with single-hop edit prompts, IFMET employs multi-hop edit prompts to precisely locate knowledge requiring modification and subsequently edits the deeper MLP layers. Comprehensive experimental evaluation demonstrates that IFMET substantially outperforms other methods.

update after rebuttal

My concerns have been addressed, and I maintain my score.

**Claims And Evidence:**

The claims made in the submission are supported by clear and convincing evidence

**Essential References Not Discussed:**

No

**Experimental Designs Or Analyses:**

The validity and rationality of the experimental design and analysis of the IFMET method are mainly demonstrated through experiments on the answer accuracy of the original answers and the edited answers, as well as comparative experiments across multiple models and ablation studies.

**Methods And Evaluation Criteria:**

Experimental results show that IFMET exhibits superior performance.

**Other Comments Or Suggestions:**

Refer to weakness.

**Other Strengths And Weaknesses:**

Strengths:
Clear Structure and Precise Expression: The quality of the writing is good. Complex concepts and intermediate solutions are effectively expressed through well-chosen examples and informative visualizations.

Novel Insight: The paper highlights the limitations of current knowledge editing methods and provides valuable guidance for future research. The analysis of reasoning mechanisms in knowledge editing scenarios and the discovery of implicit subject information in intermediate layers contribute significantly to our understanding of LLM reasoning.

Strong Theoretical Support of Method: The authors have constructed a rigorous logical chain through extensive exploratory experiments and theoretical derivations, making the logical foundation of the proposed methodology, IFMET, both solid and well-supported, with high interpretability.

Experiments: IFMET improves both original and new answers, has been tested across various models, and demonstrates its effectiveness. The authors also provide a detailed discussion of generalizability.

Weaknesses:
The IFMET framework proposed is based on existing knowledge editing methods, but seems to have been experimented on PMET, hoping to be able to extend the framework to other editing methods (such as MEMIT) to demonstrate the generalization of the method itself.

**Questions For Authors:**

Refer to weakness.

**Relation To Broader Scientific Literature:**

Knowledge editing is an important research field, and the analysis of reasoning mechanisms and the improvement of methods in knowledge editing scenarios are very valuable.

**Theoretical Claims:**

Yes, The theories of logitlens and causal intervention methods used in mechanism exploration have been checked.

---

> ### Author Rebuttal · Authors · 2025-04-01
>
> Thank you very much for your recognition of our work and for your suggestion to extend the framework to other editing methods (such as **MEMIT**) to demonstrate the generalization of the method itself. We conducted detailed experiments to verify this point. Due to time constraints, we tested it on the ablation set (a subset of 500 items consistent with the ablation experiment). The experimental results on **MEMIT** are as follows
>
>
> | Editor | Multi-hop | Efficacy |  Specificity | Paraphrase
> | -------- | -------- | -------- | -------- |  -------- |
> | MEMIT     | 15.6     | 93.0    | 76.6| 76.4 |
> | IFMET(with MEMIT) | 24.2 | 100 | 67.2 | 82.2
>
> It can be seen that using **MEMIT** as the baseline and using **PMET** as the baseline (Table 7) have similar trends in the four metrics of **Multi-hop Acc, Efficacy, Specificity, and Paraphrase**, which proves that our framework is generalizable on different editing baselines.

---

### Official Review · Reviewer_JTzm · 2025-03-12

**Overall Recommendation:** 3

**Summary:**

This paper introduces a method called IFMET to perform multi-hop factual edits to language models. It first conducts an analysis of LM factual recall in the presence of multi-hop edits and finds that LMs integrate hops beyond the second in deeper MLP layers, compared to single-hop facts which are retrieved in shallow MLP layers. It then introduces IFMET, a two-stage edit process that propagates to multi-hop facts by first editing the first-hop fact at shallow MLP layers, then editing the subsequent-hop facts at deeper MLP layers. It uses a search-then-edit process in both stages, first locating where the nth-hop fact is stored using a n-hop edit prompt, then editing the MLP at that location to store the new fact. Experimental results show that IFMET significantly improves performance on multi-hop factual recall tasks compared to previous locate-then-edit methods. Ablation studies further confirm the importance of both stages of editing, the use of multi-hop prompts, and the targeting of deeper MLP layers for effective multi-hop knowledge editing.

**Claims And Evidence:**

Examining the claims made and the evidence presented
1. Multihop facts accumulate at the last token position compared to single-hop facts. This is supported by logitlens finding that the 2nd-hop fact subject peaks in the middle layers of the last token, while the single-hop fact subjects do not peak at all at the last token. It is also supported by a causal analysis: modifying the representation at that layer (/at those MLPs) to erase the subject of the 2nd-hop fact decreases the probably of the correct answer, compared to erasing a random token.
    1. Figure 2 shows that the final answer probability actually decreases in the last few layers (beyond layer 25) which seems questionable to me. I expect the final answer probability to peak at the last layer?
    2. The effect size is small for the causal analyses: the probability of the final token decreases by only 2.5% when the 2nd-hop fact is erased in the residual stream, and on the order of 1e-6 when it is erased in the MLP. This makes me wonder if the information is more distributed than the authors claim here. It's also questionable to me why, in the MLP version of this analysis, the effect is measured at the *residual stream of the layer*, rather than the final output...
    3. While the results *do* show the final token has more information about the 2nd-hop fact compared to the 1st-hop fact, and that removing this information has a small degree of causal effect on the final answer compared to removing a random token, it's unclear whether this information is actually localized to the final token rather than an intermediate token. The authors should do some comparison on effect sizes through different positions in the prompt.
    4. Furthermore, the claim that multi-hop facts are accumulated into/applied at the MLP layers should be checked, especially because of the relatively small effect size. Other layers, like attention, should be compared against.
    5. Experiments are run on 2-hop facts but are used to generalize n-hop facts. Are similar results observed in n-hop facts?
2. An implicit assumption is that the multi-hop fact *should* change if a fact is edited. It may be good to contextualize when this would be desired: for example, if the capital of Spain changes to Hartford, then it could be possible that we are in a counterfactual world where Spain and Connecticut switched names, so Pablo Picasso was actually born in "Connecticut" which still has capital "Madrid". Of course, it is more likely is that the capital of Spain changed its name.
3. IFMET outperforms prior methods at multi-hop edits. This seems well-supported by performance improvements on multi-hop editing facts compared to baselines, especially PMET, and ablation studies show the necessity of each part of the method.

**Essential References Not Discussed:**

N/A

**Experimental Designs Or Analyses:**

The experimental designs generally seem sound. The authors perform evaluation against baselines on MQuAKE and compare results based on # of edits and # of hops. There is also comprehensive ablation analysis on each component of the IFMET method, showing that both stages are necessary, and using multi-hop prompts at deeper MLP layers are necessary.

**Methods And Evaluation Criteria:**

The method generally makes sense: MLP layers are modified for both the single-hop and multi-hop versions of prompts. MQuAKE-3K, a multi-hop knowledge editing benchmark, was used for evaluation. It should be clarified whether the same multi-hop prompts were used during editing as during evaluation, e.g. it seems important to validate that the edits generalize to when the fact is present alongside in a multi-hop chain with facts not seen during editing (e.g. if we change Spain's capital to "Hartford", then if its multihop edit was done using the prompt "Pablo Picasso's country's capital is Hartford", then it should be evaluated on the prompt "the capital of the 4th most populous country in Europe is Hartford" or even "Hartford is in Europe"). In general, IFMET appears to be an effective method for editing multi-hop facts, and does not seem to negatively impact single-hop facts. However, there are several limitations of this method compared to prior work, especially because more edits need to be made, for example:
1. Negatively impacting specificity (Table 8) indicating that other unrelated facts are being affected by these edits
2. Being more expensive because of the need to make more edits (Table 9)
3. Assuming access to a knowledge base (e.g. WikiData/SPARQL) in order to construct multihop prompts to use for editing, which prior methods do not.

**Other Comments Or Suggestions:**

N/A

**Other Strengths And Weaknesses:**

See strengths and weaknesses above.

In addition, some clarity suggestions below:
1. Important methodological details about IFMET is allocated to the appendix, as well as context for understanding the metrics used in Table 4. These should be moved up, and perhaps some details about the ablations and generalizability can be allocated to the appendix.
2. The paper should -- early on -- clarify whether what matter is the *total* number of hops in the edit, or how many hops deep the *edited fact* is. For example, it is my understanding that editing the first fact in the chain is akin to a "single-hop" edit (and is stored in shallow MLP layers), while editing the nth fact in the chain is akin to a multi-hop edit (and is stored in deep MLP layers). So it doesn't matter how long the multi-hop chain is in total, just the (absolute?) position of the edited fact in the chain. This should be clarified early on.

**Questions For Authors:**

1. How are 3/4-hop facts stored? Are they all stuck in the same MLP layer past the first hop? Or is there a smooth correlation of depth and position in the edit chain?
2. "Even in the pre-type tasks, where previous methods are relatively more proficient, IFMET achieves a significant improvement." (L385 right column) Why is this the case?

**Relation To Broader Scientific Literature:**

The authors compare against essential prior work on model weight editing like MEND, ROME, MEMIT, PMET, as well as finetuning. They show improvements against these baselines on multi-hop knowledge edits on an established benchmark, MQuaKE. They make an interesting discovery that multi-hop information is stored in later layers than single-hop information, and accumulated at the last token, which builds upon prior understanding of where single-hop knowledge is stored.

**Theoretical Claims:**

No theoretical claims in this paper.

---

> ### Author Rebuttal · Authors · 2025-04-01
>
> > Response to Q1.1 in Claims And Evidence.
>
> In our actual experiments, we conducted a simple exploration, and found that the probability of articles, e.g. *the*, *a* will increase significantly in the last few layers, which will squeeze the absolute probability of the answer to a certain extent. But the answer token is still in the position with the highest probability, so it will be output in the last layer.
>
> > Response to Q1.2 & Q1.3 & Q1.4 in Claims And Evidence about the causal intervention experiments.
>
> First of all, we need to state that we reported the absolute probability change in the paper. This was our oversight. Based on your and reviewer ixiZ’s suggestions, we revised the causal intervention method in the paper. We re-conducted a detailed intervention experiment with a window size = 5 to explore **the the role of layer hidden state and components like MLP and ATTN at different positions of prompt** with the decrease of probability in the last prediction layer as the causal effect *IE*.  The experimental results are as follows:
> 1. Prompt position: We compared last subject token position (which is also generally considered important), the last token of first hop and the last token position. **Any intervention effect at the last subject token position (<3%) and last token of fist hop (<2.5%) was weak**  at the layer level or at the attention head and mlp levels. The response at the last token position was significantly stronger with **largest response over 12%** in deep layers.
> 2. Layer intervention: We reached a consistent conclusion that the highest decay occurred in the **16-21th** layer, whose largest effect about **12%** on prob. We compared the strength of the average causal effect mentioned in figure 2 and 3 of the paper[1]. The intervention effect ranged from about 2.5% to 15%. We believe that such strength on prob and logit is acceptable.
> 3. Component: We compared the effect of intervening the input of the attention head and the MLP on answer prediction. **The effect of the MLP is significantly better than that of the ATTN head, generally reaching more than 12% in the deep 18th-24th layers**, while the ATTN is less than 3% over all layers.
> 4. In general, the results of the intervention experiment are consistent with the phenomenon observed in Figure 2. The implicit subject information plays a causal role in the generation of the final answer through the deep MLP.
>
> [1] Locating and Editing Factual Associations in GPT
> > Response to Q2 in Questions For Authors about the performance of pre-type tasks.
>
> Sorry for that our definition of **pre and post type** in the Table 4 is ambiguous. In the paper, we have added the cases of editing the first n consecutive hops into the **pre** category, such as editing the first and second hops in 3 hops. But as you said, the important insight is to divide according to whether only the first hop is edited, so we reclassified the results into **pre and post type**, corresponding to the modification of only the first hop (i.e. explicit hop) or including n>1 hops (implicit hop) in correspondence with our conclusions in the exploration section and esperiment results in Table 2. After the new classification, there are 426 pre-type cases and 2574 post-type cases, the results in Table 4 under the new classification are as follows:
>
>
> | Editor |  Average↑(Edited Answer) | Pre↑ | Post↑ |  Average↓ (Undited Answer)| Pre↓ | Post↓|
> | -------- | -------- | -------- | -------- | -------- | -------- | -------- |
> | Base     | 7.70     |            11.97    | 6.99 | 39.63 | 34.04 | 40.56 |
> | Base+CoT | 6.83     |            8.22     | 6.60 | 42.83 | 41.31 | 43.08 |
> | PMET     | 11.17    |            39.20    | 6.53 | 29.95 | **10.09** | 33.26 |
> | PMET+CoT | 17.04    |            43.66    | 12.63| 29.35 | 12.67 | 32.13 |
> | IFMET    | 23.04    |            38.03    | 20.55| 23.08 | 11.27 | 25.02 |
> | IFMET+CoT| **31.01**    |            **43.66**    | **28.90**| **21.32** | 10.80 | **23.08**|
>
> It can be seen that the previous method itself achieved significant results on pre type, as we expected, and our method is basically on par with it (this is predictable because the knowledge in the early MLP is fully updated in both cases). On post type, our method shows obvious superiority. This further confirms our findings in the paper. We will modify this in the paper, which will greatly improve the consistency and clarity of our article.
>
> > Response to Q1 in Questions For Authors & Q1.5 in Claims And Evidence about n-hop
>
> The reason why we temporarily conducted experiments on 2-hop is partly because we followed the basic settings of previous work; on the other hand, the performance of the model for three-hop and four-hop in the case of cloze-prompt (being able to directly answer the final answer in the form of the highest probability token) is not ideal. Due to the limitations of data volume, existing models, and computing resources, we hope to explore this in future work.

---

### Official Review · Reviewer_iMRZ · 2025-03-16

**Overall Recommendation:** 5

**Summary:**

This paper identifies significant limitations in the existing knowledge editing methods based on the "locate-then-edit" paradigm, particularly when applied to multi-hop factual recall tasks. To explore these limitations, the authors first employed the LogitLens technique and discovered that multi-hop queries tend to cluster implicit subject information at the last token position, while single-hop queries do not. Next, they used causal inference techniques to confirm that implicit subjects impact the final inference results and pinpointed that the key component involved in this process is the deep MLP layer. Based on these findings, the authors analyzed existing knowledge editing methods and found that most of them overlook the editing of deep MLP layers, focusing primarily on shallow MLP layers, beacuse of the use of single-hop edit prompt. This oversight leads to limitations in the performance of multi-hop factual recall tasks. To address this issue, the authors propose IFMET, a method that balances the editing of both shallow and deep MLP layers with multi-hop edit pormpts, thereby enhancing the capability of the "locate-then-edit" paradigm for multi-hop tasks.

## update after rebuttal

My concerns are addressed, and I maintain my positive score.

**Claims And Evidence:**

The claims made in the submission are supported by clear and convincing evidence

**Essential References Not Discussed:**

No.

**Experimental Designs Or Analyses:**

Yes, based on the content of the paper, the validity and soundness of the experimental design and analysis are primarily demonstrated through comparative experiments across multiple models and ablation studies. The experimental design addresses several key aspects: it identifies the key differences in the mechanisms the model uses for reasoning in single-hop versus multi-hop fact recall tasks. The paper evaluates multi-hop tasks across various models and conducts a well-designed set of ablation experiments to demonstrate the effectiveness of the proposed method. The design and analysis methods are appropriate and effective, with no obvious issues or flaws found. All experimental and analytical steps have been thoroughly validated.

**Methods And Evaluation Criteria:**

Yes, The proposed method addresses the issue where post-edited models struggle with multi-hop factual recall tasks, particularly those that involve newly edited knowledge. Through extensive comparative experiments and ablation studies across various models, the paper demonstrates the effectiveness of the proposed approach.

**Other Comments Or Suggestions:**

See weaknesses

**Other Strengths And Weaknesses:**

**Strengths**
- Empirical analysis: The paper provides a solid empirical analysis, validating two hypotheses about how large language models (LLMs) process multi-hop queries compared to single-hop ones. Using interpretability tools like Logitlens and causal intervention, the authors identify key insights that enhance our understanding of LLM reasoning and model interpretability.
- Effective approach: The IFMET solution is well-founded and effectively addresses multi-hop reasoning challenges, showing notable improvements in performance, with potential real-world applications.
- Thorough experimentation: The experimentation is thorough, with ablation studies and comparisons to existing methods, which effectively demonstrate the approach's superiority in multi-hop factual recall tasks.

**Weaknesses**
- The paper's formatting needs improvement, and the content in the experimental section is relatively limited. Some important results, such as the ablation experiments on LLaMA2, are presented in the appendix. It is hoped that adjustments can be made in the camera-ready version.
- I strongly recommend that the authors include a discussion of in-context editing approaches in relevant sections, such as Related Work or Appendix.

**Questions For Authors:**

1.	How is the dimension of K0 determined in Equation 3?
2.	Is the performance of the methods in Table 3 related to the number of edits? As shown in the table, the performance of 3-edit is worse than both 2-edit and 4-edit.

**Relation To Broader Scientific Literature:**

Knowledge editing is an active research area in natural language processing. IFMET advances this field by introducing a two-stage editing strategy (shallow and deep editing), especially addressing the challenges in multi-hop reasoning tasks. IFMET also modifies knowledge based on interpretability guidance, which aligns with recent research on model interpretability and reasoning transparency.

**Theoretical Claims:**

Yes, the paper provides a closed-form solution to the incremental weight, and this section includes theoretical claims. The solution process is correct and free from issues.

---

> ### Author Rebuttal · Authors · 2025-04-01
>
> Thank you very much for your recognition of our work. Here is our response.
> > Response to Q1:How is the dimension of K0 determined in Equation 3?
>
> K0 represents the knowledge we try to preserve when modifying specific fact. Throughout the calculation process, we do not calculate K0 separately, but follow the practice in Rome[1], treating $C = KK^T$ in Equation 3 as a constant that we pre-cache by estimating the uncentered covariance of k from a sample of Wikipedia text. Our second moment statistics C ∝ E[$kk^T$] are computed using 100,000 samples of hidden states k computed from tokens sampled from all Wikipedia text in-context.
>
> > Response to Q2: Is the performance of the methods in Table 3 related to the number of edits? As shown in the table, the performance of 3-edit is worse than both 2-edit and 4-edit.
>
> We believe that the performance of answering multi-hop questions after the model implements knowledge editing is usually inversely proportional to the number of edits. However, in the actual implementation of the KE method, it is also related to the quality of the prompt used for editing. We checked the prompts used in the 3-edit part and found that the confidence of the model when answering based on them was relatively low, which means that the model is more likely to be biased when recalling, which means that the multi-hop prompt in 3-edit cases may not be optimal, but we believe that constructing a better multi-hop prompt can be left for future exploration. This article only represents the lower limit performance of the proposed framework.
>
> [1] Locating and Editing Factual Associations in GPT

---

> > ### Comment · Reviewer_iMRZ · 2025-04-05
> >
> > Thank you for your further response. My concerns have been well addressed.

---

### Decision · Program_Chairs · 2025-05-01

**Decision:**

Accept (poster)

**Comment:**

This paper studies multi-hop knowledge editing, using mechanistic interpretability methods to identify failures of existing editors in multi-hop knowledge editing. They then use these insights to develop a new editing method that makes multi-hop knowledge edits by conducting edits in two stages. Their approach outperforms the state-of-the-art on multi-hop knowledge editing tasks.

Reviewers ultimately felt the strengths of this work outweigh the weaknesses. They particularly appreciated the work's clarity, useful insights, problem importance, empirical analysis, and resultant performance of the method.